# Multimodal analysis of methylomics and fragmentomics in plasma cell-free DNA for multi-cancer early detection and localization

Van Thien Chi Nguyen[1,2†], Trong Hieu Nguyen[1,2†], Nhu Nhat Tan Doan[1,2], Thi Mong Quynh Pham[1,2], Giang Thi Huong Nguyen[1,2], Thanh Dat Nguyen[1,2], Thuy Thi Thu Tran[1,2], Duy Long Vo[3], Thanh Hai Phan[4], Thanh Xuan Jasmine[4], Van Chu Nguyen[5,6], Huu Thinh Nguyen[3], Trieu Vu Nguyen[7], Thi Hue Hanh Nguyen[1,2], Le Anh Khoa Huynh[1,8], Trung Hieu Tran[1,2], Quang Thong Dang[3], Thuy Nguyen Doan[3], Anh Minh Tran[3], Viet Hai Nguyen[3], Vu Tuan Anh Nguyen[3], Le Minh Quoc Ho[3], Quang Dat Tran[3], Thi Thu Thuy Pham[4], Tan Dat Ho[4], Bao Toan Nguyen[4], Thanh Nhan Vo Nguyen[4], Thanh Dang Nguyen[4], Dung Thai Bieu Phu[4], Boi Hoan Huu Phan[4], Thi Loan Vo[4], Thi Huong Thoang Nai[4], Thuy Trang Tran[4], My Hoang Truong[4], Ngan Chau Tran[4], Trung Kien Le[3], Thanh Huong Thi Tran[5,6], Minh Long Duong[5,6], Hoai Phuong Thi Bach[5,6], Van Vu Kim[5,6], The Anh Pham[5,6], Duc Huy Tran[3], Trinh Ngoc An Le[3], Truong Vinh Ngoc Pham[3], Minh Triet Le[3], Dac Ho Vo[1,2], Thi Minh Thu Tran[1,2], Minh Nguyen Nguyen[1,2], Thi Tuong Vi Van[1,2], Anh Nhu Nguyen[1,2], Thi Trang Tran[1,2], Vu Uyen Tran[1,2], Minh Phong Le[1,2], Thi Thanh Do[1,2], Thi Van Phan[1,2], Hong-Dang Luu Nguyen[1,2], Duy Sinh Nguyen[1,2], Van Thinh Cao[9], Thanh-Thuy Thi Do[2], Dinh Kiet Truong[2], Hung Sang Tang[1,2], Hoa Giang[1,2], Hoai-Nghia Nguyen[1,2], Minh-Duy Phan[1,2]*, Le Son Tran[1,2]*

[1]Gene Solutions, Ho Chi Minh City, Viet Nam; [2]Medical Genetics Institute, Ho Chi Minh City, Viet Nam; [3]University Medical Center, Ho Chi Minh City, Viet Nam; [4]MEDIC Medical Center, Ho Chi Minh City, Viet Nam; [5]National Cancer Hospital, Hanoi, Viet Nam; [6]Hanoi Medical University, Hanoi, Viet Nam; [7]Thu Duc City Hospital, Ho Chi Minh City, Viet Nam; [8]Department of Biostatistics, Virginia Commonwealth University, School of Medicine, Richmond, United States; [9]Pham Ngoc Thach University of Medicine, Ho Chi Minh City, Viet Nam

*For correspondence:
pmduy@yahoo.com (M-DP);
leson1808@gmail.com (LST)

†These authors contributed
equally to this work

**Abstract** Despite their promise, circulating tumor DNA (ctDNA)-based assays for multi-cancer early detection face challenges in test performance, due mostly to the limited abundance of ctDNA and its inherent variability. To address these challenges, published assays to date demanded a very high-depth sequencing, resulting in an elevated price of test. Herein, we developed a multimodal assay called SPOT-MAS (screening for the presence of tumor by methylation and size) to simultaneously profile methylomics, fragmentomics, copy number, and end motifs in a single workflow using targeted and shallow genome-wide sequencing (~0.55×) of cell-free DNA. We applied SPOT-MAS to 738 non-metastatic patients with breast, colorectal, gastric, lung, and liver cancer, and 1550 healthy controls. We then employed machine learning to extract multiple cancer and tissue-specific signatures for detecting and locating cancer. SPOT-MAS successfully detected the five cancer types with a sensitivity of 72.4% at 97.0% specificity. The sensitivities for detecting early-stage cancers were 73.9% and 62.3% for stages I and II, respectively, increasing to 88.3% for non-metastatic stage

IIIA. For tumor-of-origin, our assay achieved an accuracy of 0.7. Our study demonstrates comparable performance to other ctDNA-based assays while requiring significantly lower sequencing depth, making it economically feasible for population-wide screening.

## eLife assessment

This study provides insights into the early detection of malignancies with noninvasive methods by developing a framework, which assesses methylation, CNA, and other genomic features. They established a **solid** model in discriminating malignancies from healthy controls, as well as the ability to distinguish tumor of origin. This **important** study will demonstrate its practical impacts in the clinic and other researchers of the field.

## Introduction

The incidence of cancer-related morbidity and mortality is rapidly increasing globally, and accounted for nearly one fifth of all deaths in 2020 (*Sung et al., 2021*). High-cost treatment is a significant financial burden for cancer patients, with almost 286 billion dollars in 2021 and an increase of 8.2% – 581 billion dollars in 2030. In Vietnam, GLOBOCAN 2020 reported over 182,500 newly diagnosed cases and 122,690 cancer-related deaths (*Sung et al., 2021*). Among these, liver (14.5%), lung (14.4%), breast (11.8%), gastric (9.8%), and colorectal cancer (9%) are the five most common types. Up to 80% of cancer patients in Vietnam were diagnosed at stage III or stage IV, resulting in a high rate of 1 year mortality (25%) and a low 5-year survival rate compared to other countries (*Pham et al., 2019*). Diagnostic delays are associated with a lower chance of survival, greater treatment-associated problems, and higher costs (*Hawkes, 2019*). Cancer detection at earlier stages can improve the opportunity to control cancer progression, increase the patient survival rate, and lower medical expenses (*Kakushadze et al., 2017*).

Although currently guided screening tests have each been shown to provide better treatment outcomes and reduce cancer mortality, some of them are invasive, thus having low accessibility. Importantly, most of them are single cancer screening tests, which may result in high false positive rates when used sequentially (*Sasieni et al., 2023*). Multi-cancer early detection (MCED) tests can potentially overcome these challenges by simultaneously detecting multiple cancer types from a single test (*Liu et al., 2020*). Liquid biopsy, an emerging non-invasive approach for MCED, can capture a wide range of tumor features, including cell-free DNA (cfDNA), circulating tumor DNA (ctDNA), exosomes, proteins, mRNA, and metabolites (*Li et al., 2018*; *Nguyen et al., 2022b*). Among them, ctDNA has become a promising biomarker for detecting early-stage cancers because it is a carrier of genetic and epigenetic modifications from cancer-derived DNA (*Gao et al., 2022*). Indeed, ctDNA detection has demonstrated several advantages in non-invasive diagnostic, prognostic, and monitoring of cancer patients during and after treatment (*Pascual et al., 2022*; *Nguyen et al., 2020*). Furthermore, ctDNA carrying tumor-specific alterations could be used to identify the corresponding unknown primary cancer and tumor localization.

In recent years, there has been considerable interest in exploring the potential of ctDNA alterations for early detection of cancer (*Nguyen et al., 2020*; *Moser et al., 2023*). One such approach is the PanSeer test, which uses 477 differentially methylated regions (DMRs) in ctDNA to detect five different types of cancer up to 4 years prior to conventional diagnosis (*Im et al., 2021*). The DELFI assay employs a genome-wide analysis of ctDNA fragment profiles to increase sensitivity in early detection (*Zhou et al., 2022b*). Recently, the Galleri test has emerged as a multi-cancer detection assay that analyzes more than 100,000 methylation regions in the genome to detect over 50 cancer types and localize the tumor site (*Cohen et al., 2018*).

Despite their great potential, there remain several challenges that these assays must solve to deliver accessible and reliable clinical adoption for the large population, including the low fraction of ctDNA in the blood of early-stage cancer patients, the heterogeneity of ctDNA signatures from diverse cancer types, subtypes and stages (*Moser et al., 2023*), and the high sequencing depth required. To address these challenges, recent studies have focused on multi-analyte approach – combining genomic and nongenomic features such as methylomics and fragmentomics to increase the detection of ctDNA and accuracy for tissue of origin (TOO) identification (*Moser et al., 2023*; *Im*

*et al., 2021*; *Zhou et al., 2022b*). Advances in multimodal analysis approaches have led to the development of powerful screening tests that enable high sensitivity and cost-effectiveness. For example, CancerSEEK uses a combined approach of protein biomarkers and genetic alterations to detect and locate the presence of eight types of cancers (*Cohen et al., 2018*). In this assay, cancer-associated serum proteins play a complementary role in tumor localization as cfDNA mutations are not tissue specific. However, detecting both protein and genetic biomarkers are time-consuming and costly. Thus, the development of future MCED tests should endeavor to deliver a screening approach with high sensitivity, specificity, and TOO identification at cost-effective price to provide better clinical outcomes and treatment opportunities for all cancer patients.

In an effort to address the challenges of early cancer detection, we have developed a multimodal approach called SPOT-MAS (screening for the presence of tumor by DNA methylation and size). This assay was previously applied to cohorts of colorectal (*Nguyen et al., 2022a*) and breast cancer patients (*Pham et al., 2023*) and demonstrated ability for early detection of these cancers at high sensitivity across different cancer stages and patient age groups. In this study, we aimed to expand our multimodal approach, SPOT-MAS, to comprehensively analyze methylomics, fragmentomics, DNA copy number, and end motifs (EMs) of cfDNA and evaluate its utility to simultaneously detecting and locating cancer from a single screening test. As proof of concept, we used 2288 participants, including 738 non-metastatic patients and 1550 healthy controls, to train and fully validate this approach on five commonly diagnosed cancers, including breast, gastric, lung, colorectal, and liver cancer. Our findings demonstrate that the multimodal approach of SPOT-MAS enables profiling of multiple ctDNA signatures across the entire genome at low sequencing depth to detect five different cancer types in their early stages. Beyond detecting the presence of cancer signals, our assay was able to predict the tumor location, which is important for clinicians to fast-track the follow-up diagnostic and guide necessary treatment. Thus, SPOT-MAS has the potential to become a universal, simple, and cost-effective approach for early multi-cancer detection in a large population.

## Results
### Clinical characteristics of cancer and healthy participants

This study recruited 738 patients with five common cancer types, including breast cancer (*n*=223), CRC (*n*=159), gastric cancer (*n*=98), liver cancer (*n*=122), lung cancer (*n*=136), and 1550 healthy participants (*Supplementary file 1*, Table S1). Cancer patients were diagnosed by either imaging and/or histology analysis, depending on cancer type. All cancer patients were treatment-naïve at the time of blood collection. Healthy participants had no history of cancer at the time of sample collection

**Table 1.** Summary of clinical features of 738 cancer patients and 1550 healthy controls in discovery and validation cohorts.

| Clinical features | | Discovery cohort (*N*=1575) | | | | | Validation cohort (*N*=713) | | | | |
|---|---|---|---|---|---|---|---|---|---|---|---|
| | | Cancer (*N*=499) | | Healthy (*N*=1076) | | p-Value (cancer vs healthy) | Cancer (*N*=239) | | Healthy (*N*=474) | | p-Value (cancer vs healthy) |
| | | *N* | Percentage | *N* | Percentage | | *N* | Percentage | *N* | Percentage | |
| Gender | Female | 279 | 55.9% | 599 | 55.7% | 0.9281* | 126 | 52.72% | 270 | 56.1% | 0.2818* |
| | Male | 220 | 44.1% | 477 | 44.3% | | 113 | 47.28% | 204 | 43.9% | |
| Age | Median | 58 | | 47 | | <0.0001† | 59 | | 48 | | <0.0001† |
| | Min | 25 | | 18 | | | 28 | | 19 | | |
| | Max | 97 | | 84 | | | 92 | | 85 | | |
| Stage | I | 52 | 10.4% | | | | 23 | 9.6% | | | 0.4947* |
| | II | 169 | 33.9% | | | | 69 | 28.9% | | | |
| | IIIA | 150 | 30.1% | | | | 77 | 32.2% | | | |
| | Non-metastasis with unknown staging information | 128 | 25.7% | | | | 70 | 29.3% | | | |

*p-Values from Chi-square test.
†p-Values from Mann-Whitney test.

and remained cancer-free at the 6- and 12-month follow-ups. Cancer patients and healthy participants were randomly assigned to the discovery and validation cohorts (*Table 1* and *Supplementary file 1*, Table S2). The discovery cohort was used to profile multiple cancer- and tissue-specific signatures and to construct machine learning algorithm while the validation cohort was used solely to external evaluation of the performance of machine learning models.

The discovery cohort comprised of 499 cancer patients (156 breast, 106 CRC, 67 gastric, 77 liver, and 93 lung, *Supplementary file 1*, Table S1) and 1076 healthy participants. The cancer group had a median age of 58 (range 25–97, *Table 1*) and consisted of 279 females and 220 males. The discovery healthy group consisted of 599 females and 477 males, with a median age of 47 (range 18–84, *Table 1*). In the discovery cohort, gender ratios were similar between cancer and healthy control groups, whereas cancer patients were older than controls (p<0.0001, Mann-Whitney test, *Table 1*). Of the cancer patients, 10.4% were at stage I, 33.9% were at stage II, and 30.1% were at non-metastatic stage IIIA. Staging information was not available for 25.7% of cancer patients, who were confirmed by specialized clinicians to have non-metastatic tumors (*Table 1*).

The validation cohort consisted of 239 cancer patients (67 breast, 53 CRC, 31 gastric, 45 liver, and 43 lung, *Supplementary file 1*, Table S1) and 474 healthy participants (*Table 1*). Consistent with the discovery cohort, the gender distribution was comparable between the cancer and healthy control groups, and the cancer group was older than the control group, with a median age of 59 and 48 years of age, respectively (p<0.0001, Mann-Whitney test, *Table 1*). The percentage of cancer patients with each stage was similar to that of the discovery cohort, with 9.6% at stage I, 28.9% at stage II, and 32.2% at stage IIIA. Staging information was unavailable for 29.3% of non-metastatic cancer patients (*Table 1*).

## The multimodal SPOT-MAS assay for multi-cancer and TOO detection

In our recent study of SPOT-MAS, we have demonstrated that the integration of ctDNA methylation and fragmentomic features can significantly improve the early detection of colorectal cancer (*Nguyen et al., 2022a*) and breast cancer (*Pham et al., 2023*). Here, we expanded the breadth of ctDNA analyses by adding two sets of features including DNA copy number and EM into SPOT-MAS to maximize cancer detection rate and identify TOO. Briefly, a novel and cost-effective workflow of SPOT-MAS was developed involving three main steps (*Figure 1*). In step 1, cfDNA was isolated from peripheral blood and subjected to bisulfite conversion and adapter ligation to create a single whole-genome bisulfite library of cfDNA. From this library, in step 2, a hybridization reaction was performed to collect the target capture fraction (450 cancer-specific regions), then the whole-genome fraction was retrieved by collecting the 'flow-through' and hybridizing with probes specific for adapter sequences of DNA library. Both the target capture fraction and whole-genome fraction were sequenced to the depth of ~52× and 0.55×, respectively (*Supplementary file 1*, Table S3). Data pre-processing was performed to generate five different sets of cfDNA features, including methylation changes at target regions (TM), genome-wide methylation (GWM), fragment length patterns (FLEN), copy number aberrations (CNA), and EM. In step 3, these features were used as inputs for a two-stage model to obtain prediction outcomes. Stage 1 of our model comprised of a stacking ensemble machine learning model for binary classification of cancer versus healthy. Then, the samples predicted as cancer were passed to stage 2 where graph convolutional neural network (GCNN) was adopted to predict TOO (*Figure 1*).

## Identification of DMRs in cancer patients from target capture fraction

DNA methylation is an important epigenetic signature responsible for major changes in regulating expression of cancer-associated genes by impacting the binding of transcription factors to regulatory sites and the structure of chromatin (*Yin et al., 2017*; *Buitrago et al., 2021*). Of the 450 target regions associated with cancer that were selected from public data (*Chen et al., 2020*; *Nguyen et al., 2021*; *Phan et al., 2022*), 402 regions were identified as DMRs in cancer patients when compared to healthy participants from the discovery cohort (Wilcoxon rank-sum test, p-values <0.05, *Figure 2A* and *Supplementary file 1*, Table S4). Of those, 339 (84.3%) regions were identified as hypermethylated (log$_2$ fold change [logFC] >0), and 63 (15.7%) regions as hypomethylated in cancer samples (logFC <0, *Figure 2A*). We next examined the genomic location of the 402 DMRs and found 100, 108, 107, and 87 DMRs that were mapped to promoter, exon, intron, and intergenic regions, respectively (*Figure 2B*). To understand the relationship between the differences in methylation regions and

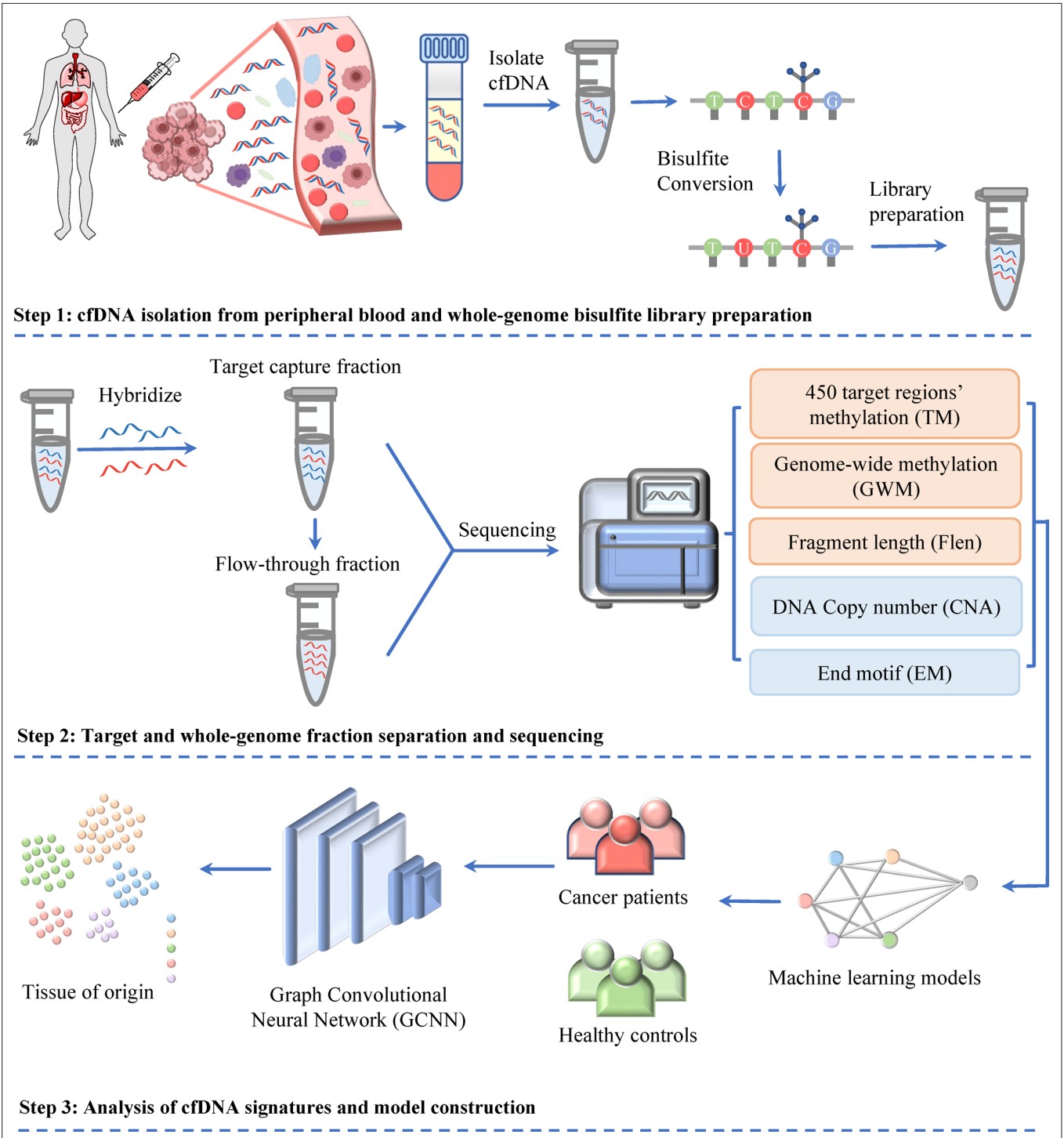

**Step 1: cfDNA isolation from peripheral blood and whole-genome bisulfite library preparation**

**Step 2: Target and whole-genome fraction separation and sequencing**

**Step 3: Analysis of cfDNA signatures and model construction**

**Figure 1.** Workflow of SPOT-MAS (screening for the presence of tumor by methylation and size) assay for multi-cancer detection and localization. There are three main steps in the SPOT-MAS assay. First, cell-free DNA (cfDNA) is isolated from peripheral blood, then treated with bisulfite conversion and adapter ligation to make whole-genome bisulfite cfDNA library. Second, whole-genome bisulfite cfDNA library is subjected to hybridization by probes specific for 450 target regions to collect the target capture fraction. The whole-genome fraction was retrieved by collecting the 'flow-through' and hybridized with probes specific for adapter sequences of DNA library. Both the target capture and whole-genome fractions were subjected to massive parallel sequencing and the resulting data were pre-processed into five different features of cfDNA: target methylation (TM), genome-wide methylation (GWM), fragment length profile (FLEN), DNA copy number (CNA), and end motif (EM). Finally, machine learning models and graph convolutional neural networks are adopted for classification of cancer status and identification tissue of origin.

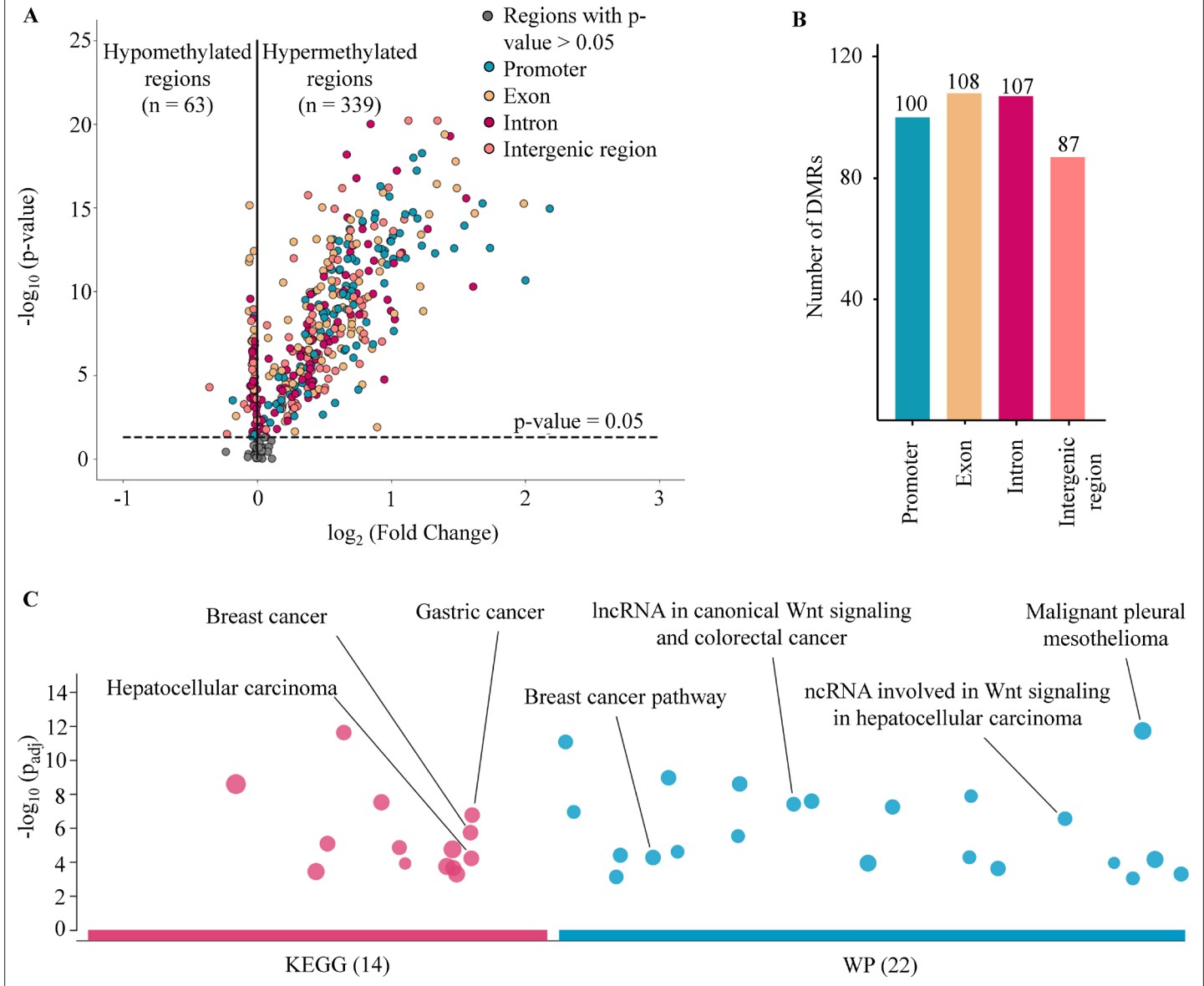

**Figure 2.** Analysis of targeted methylation in cell-free DNA (cfDNA). (**A**) Volcano plot shows log₂ fold change (logFC) and significance (-log₁₀ Benjamini-Hochberg adjusted p-value from Wilcoxon rank-sum test) of 450 target regions when comparing 499 cancer patients and 1076 healthy controls in the discovery cohort. There are 402 DMRs (p-value <0.05), color-coded by genomic locations. (**B**) Number of differentially methylated regions (DMRs) in the four genomic locations. (**C**) Kyoto Encyclopedia of Genes and Genomes (KEGG) and WikiPathway (WP) pathway enrichment analysis using g:Profiler for genes associated with the DMRs. A total of 36 pathways are enriched, suggesting a link between differences in methylation regions and tumorigenesis.

biological pathways, we performed pathway enrichment analysis using g:Profiler on hypermethylated DMRs. We detected 36 enriched pathways, including 14 from Kyoto Encyclopedia of Genes and Genomes (KEGG) and 22 from WikiPathway (WP) (*Figure 2C* and *Supplementary file 1*, Table S5). These significant pathways were known to regulate tumorigenesis of breast, gastric, hepatocellular, and colorectal cancer. Therefore, the methylation changes in the targeted regions, particularly the hypermethylated DMRs, mostly occur early in tumorigenesis and are crucial for distinguishing early-stage cancer patients from healthy individuals.

## GWM changes in cfDNA of cancer patients

In addition to site-specific hypermethylation, hypomethylation is a significant genome-wide change that has been identified in many types of cancers (*Das and Singal, 2004*; *Ehrlich, 2002*; *Hoffmann*

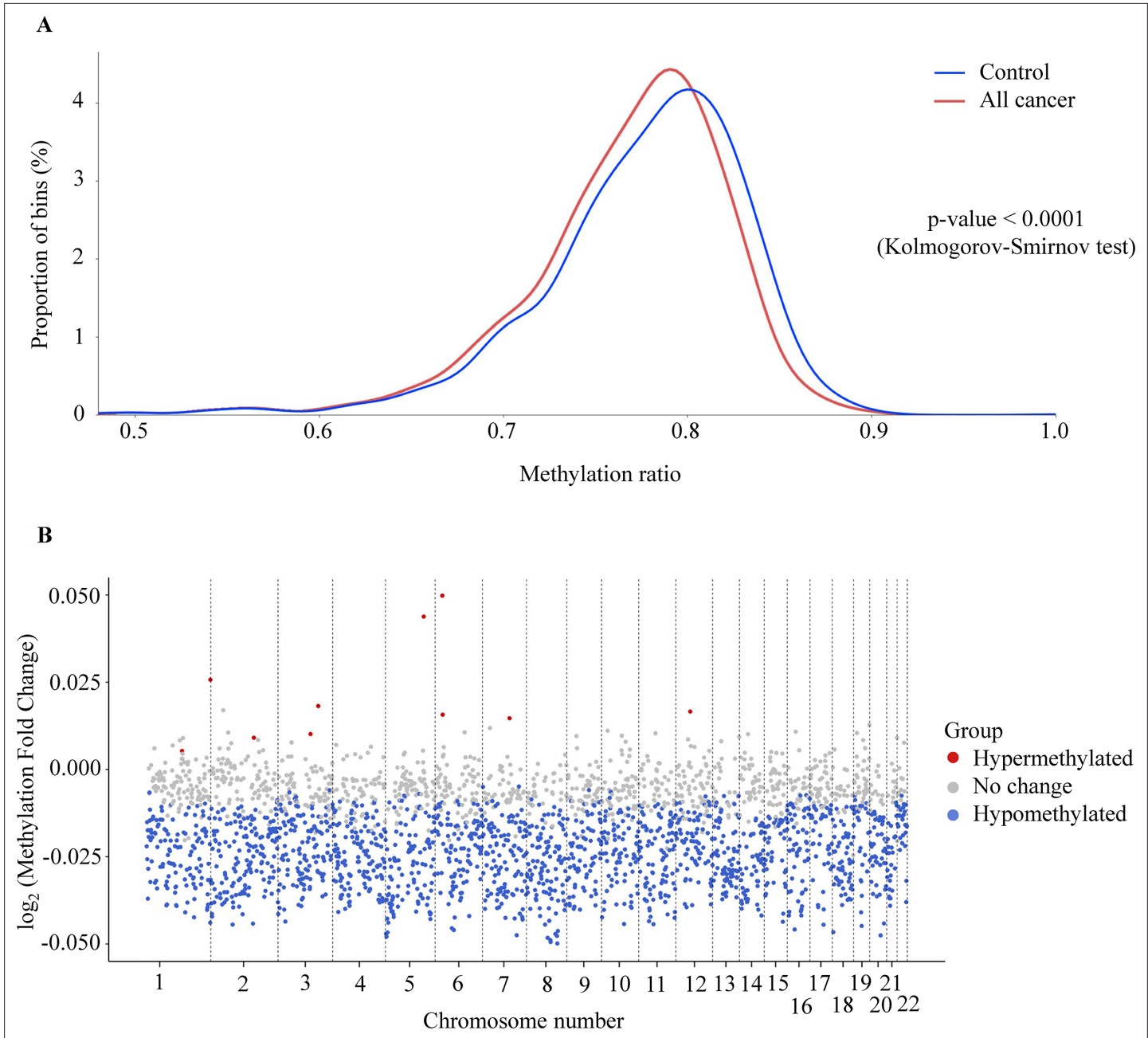

**Figure 3.** Genome-wide methylation changes in cell-free DNA (cfDNA) of cancer patients. (**A**) Density plot showing the distribution of genome-wide methylation ratio for all cancer patients (red curve, *n*=499) and healthy participants (blue curve, *n*=1076). The left-ward shift in cancer samples indicates global hypomethylation in the cancer genome (p<0.0001, two-sample Kolmogorov-Smirnov test). (**B**) Log₂ fold change of methylation ratio between cancer patients and healthy participants in each bin across 22 chromosomes. Each dot indicates a bin, identified as hypermethylated (red), hypomethylated (blue), or no significant change in methylation (gray).

*and Schulz, 2005*). To investigate the methylation changes at genome-wide level, bisulfite sequencing reads from the whole-genome fraction were mapped to the human genome, split into bins of 1 Mb (2734 bins across the genome), and the reads from each bin were used to calculate methylation ratio. As expected, we observed a left-ward shift in the distribution of methylation ratio in cancer samples compared to healthy controls, indicating global hypomethylation in the cancer genome (p<0.0001, two-sample Kolmogorov-Smirnov test, *Figure 3A*). Of these bins, we identified 1715 (62.7%) bins as significantly hypomethylated in cancer, located across 22 autosomes of the genome (*Figure 3B*, Wilcoxon rank-sum test with Benjamini-Hochberg adjusting p-value <0.05). In contrast, there were

only 10 bins identified as hypermethylated and mapped to chromosome 1, 2, 3, 5, 6, 7, and 12 in the cancer genome (*Figure 3B*). Therefore, our data confirmed the widespread hypomethylation across the genome and this would potentially serve to distinguish cancer patients from healthy controls.

## Increase DNA CNAs in cfDNA of cancer patients

Somatic CNAs in the cancer genome are associated with the initiation and progression of numerous cancers by altering transcriptional levels of both oncogenes and tumor suppressor genes (*Shao et al., 2019*). Recent studies have shown that CNAs detection could identify and quantify the fraction of ctDNA in plasma cfDNA (*Baldacchino and Grech, 2020*; *Knuutila et al., 1999*; *Dereli-Öz et al., 2011*). To examine CNAs at genome-wide scale, we used 1 Mb bin to determine the percentage of bins that showed significant copy number gains or losses between cancer and control group. We

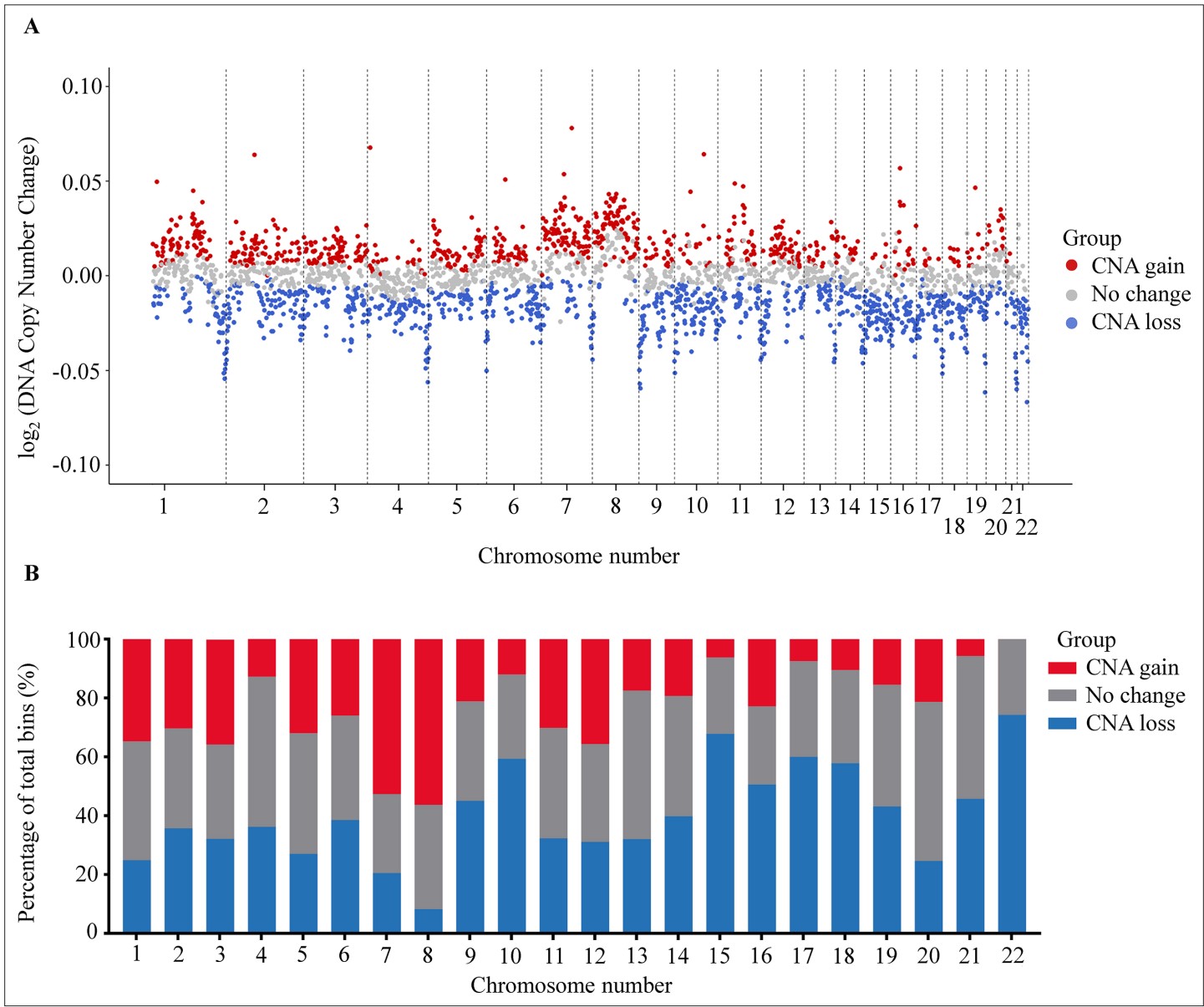

**Figure 4.** Analysis of copy number aberration (CNA) in cell-free DNA (cfDNA). (**A**) Log₂ fold change of DNA copy number in each bin across 22 autosomes between 499 cancer patients and 1076 healthy participants in the discovery cohort. Each dot represents a bin identified as gain (red), loss (blue), or no change (gray) in copy number. (**B**) Proportions of different CNA bins in each autosomes.

The online version of this article includes the following figure supplement(s) for figure 4:

**Figure supplement 1.** Association between methylation changes and copy number aberration (CNA).

identified 729 bins (27.1%) with a significant gain and 976 bins (36.3%) with a significant loss in copy number across 22 chromosomes of the cancer genome (Benjamini-Hochberg adjusting p-value <0.05, Wilcoxon rank-sum test, *Figure 4A*). We noted that chromosome 8 had the highest proportion of bins with CNA gains, while chromosome 22 showed the highest proportion of bins with CNA losses (*Figure 4B*).

It is thought that the abnormal hypomethylation at genome-wide level is linked with somatic CNA, resulting in genome instability, which is an important tumorigenic event (*Brennan and Flanagan, 2012*; *Zhang et al., 2020*; *Chan et al., 2013*). Indeed, our data showed a significant increase in levels of CNA in hypomethylated bins compared to bins with unchanged methylation (p=0.024, *Figure 4— figure supplement 1*). Consistently, bins with CNA gains showed significant decreases in methylation as compared to those with CNA losses or unchanged CNA (p<0.01, *Figure 4—figure supplement 1*). In summary, SPOT-MAS enables comprehensive profiling of both global differences in methylation and somatic CNA as individual feature types, as well as exploring their functional links during cancer initiation and development, rendering them ideal biomarkers for cancer detection.

## Fragment length analysis captured patterns of ctDNA in plasma

Several studies have shown that the fragmentation pattern of cfDNA is a non-random event mediated by apoptotic-dependent caspases and ctDNA fragments tend to be shorter than non-cancer cfDNA (*Underhill et al., 2016*; *Mouliere et al., 2018*; *Lo et al., 2021*; *Cristiano et al., 2019*; *Nguyen et al., 2023b*). One novel technical aspect of SPOT-MAS is the use of bisulfite sequencing data not only for methylation but also for fragment length analysis. Certain studies showed evidence of DNA degradation followed bisulfite treatment, possibly due to high temperature and low pH conditions of the bisulfite conversion procedure, while other showed that bisulfite sequencing affects large genomic DNA but not small size cfDNA (*Raizis et al., 1995*; *Tanaka and Okamoto, 2007*; *Kint et al., 2018*; *Ehrich et al., 2007*). Therefore, to demonstrate the use of bisulfite-treated cfDNA for fragment length analysis, we randomly selected three healthy controls and nine cancer samples to perform pair-wise comparison between bisulfite and non-bisulfite sequencing results. We observed a strong correlation between fragment length profile of non-bisulfite and bisulfite sequencing (Pearson's correlation, $R^2$ >0.9, p<0.0001, *Figure 5—figure supplement 1*) for all 12 tested samples, indicating the feasibility of using bisulfite sequencing data for cfDNA fragment length analysis. Indeed, the fragment size distributions of bisulfite-treated cfDNA in both cancer patients and control subjects showed a peak at 167 bp (*Figure 5A*), corresponding to the length of DNA wrapped around histone (~ 147 bp) plus linker regions (~ 2 × 10 bp), which was in good agreement with previous studies using non-bisulfite cfDNA (*Cristiano et al., 2019*; *Jiang et al., 2020*). Importantly, our results showed that cfDNA of cancer patients was more fragmented than that of healthy participants, with a higher frequency of fragments ≤150 bp and a lower frequency of fragment >150 bp (*Figure 5A*).

To examine whether the fragment length variation in cancer-derived cfDNA and non-cancer cfDNA could be position-dependent (*Cristiano et al., 2019*), we calculated the ratios of short (≤150 bp) to long fragments (>150 bp) across the genome in cancer patients and healthy controls. The mean ratio of short to long fragments in cancer patients was 0.29 (range 0.28–0.42), which was higher than the mean ratio of 0.27 (range 0.26–0.39) for healthy controls (*Figure 5B*). The changes of mean ratio were across 22 autosomes of the genome. Our results indicate that the SPOT-MAS technology can effectively capture differences in fragmentation patterns between cancer and healthy participants across the entire genome, making them potential biomarkers for the detection of ctDNA in plasma.

## Profile of 4-mer EMs reflecting differences between cancer and healthy cfDNA

Associated with differences in fragment length is the differences in the DNA motifs at the end of each fragment as the consequences of differential cleavage between DNA in cancer cells and normal cells during apoptosis (*Jiang et al., 2020*; *Jin et al., 2021*). Here, we calculated the frequencies of 256 4-mer EMs of cfDNA fragments and compared them between cancer patients and healthy participants. Consistent with the fragment length features, we also confirmed the correlation of EM frequency between bisulfite and non-bisulfite sequencing results of 12 randomly selected samples, suggesting that EM profiles were reserved in bisulfite-treated cfDNA (*Figure 5—figure supplement 1*). Of the 256 4-mer EMs, we detected 78 motifs with increased frequencies and 106 motifs with

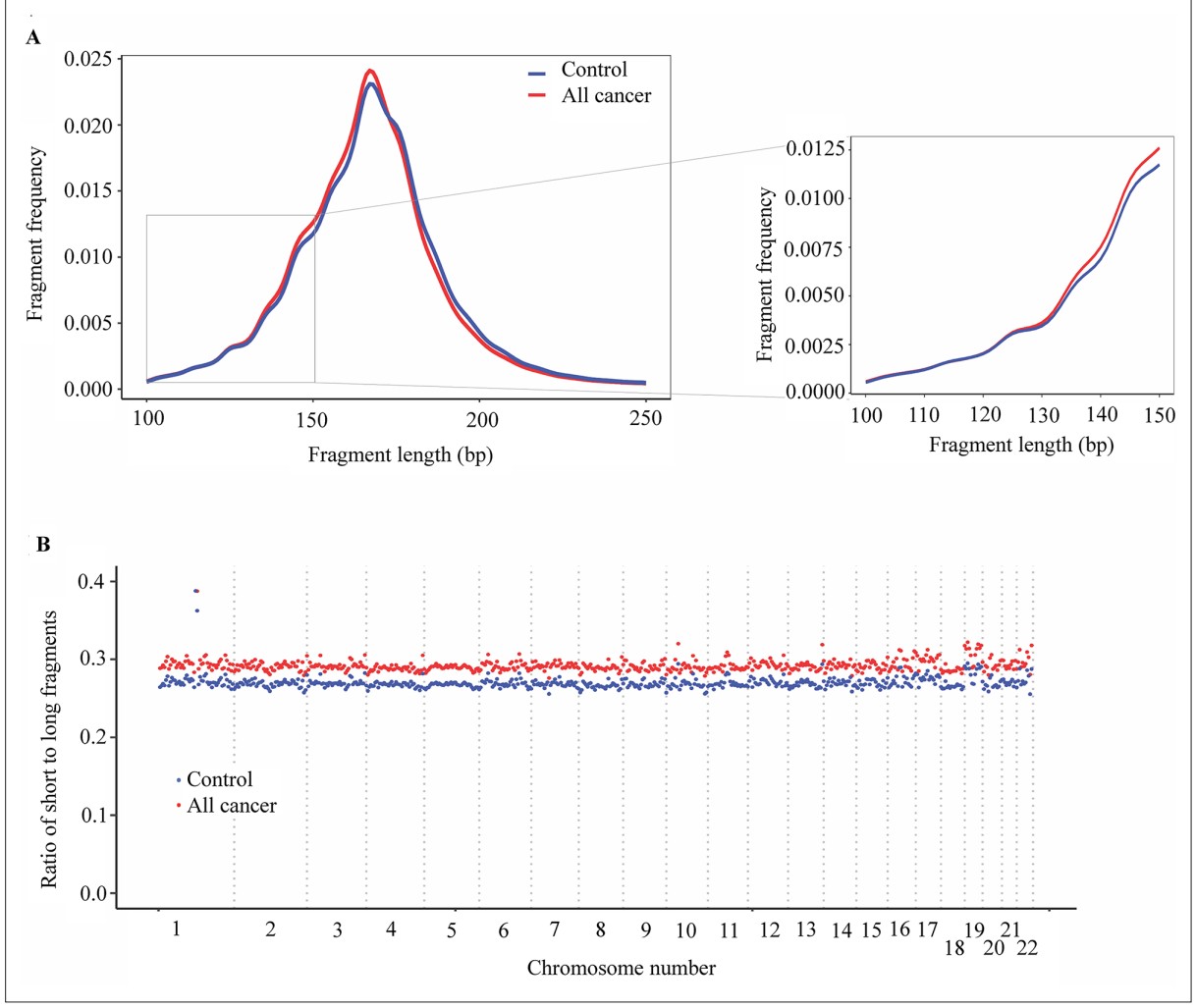

**Figure 5.** Analysis of fragment length patterns of circulating tumor DNA (ctDNA) in plasma. (**A**) Density plot of fragment length between cancer patients (red, *n*=499) and healthy participants (blue, *n*=1076) in the discovery cohort. Inset corresponds to an x-axis expansion of short fragment (<150 bp). (**B**) Ratio of short to long fragments across 22 autosomes. Each dot indicates a mean ratio for each bin in cancer patients (red) and healthy participants (blue).

The online version of this article includes the following figure supplement(s) for figure 5:

**Figure supplement 1.** Correlations between bisulfite and non-bisulfite converted data.

decreased frequencies between cancer and healthy controls (*Figure 6A* and *Supplementary file 1*, Table S6).

Interestingly, EMs beginning with cytosine (C) exhibited the highest number of EMs with significant changes of frequency in cancer samples (*Figure 6A*). *Figure 6B* shows the top 10 EMs exhibiting significant differences. Specifically, the frequencies of five motifs (CAAA, TAGA, CAGA, CAAG, and CAAT) were found to be significantly increased, while the frequencies of another five motifs (CGCT, CGCC, CGCA, GCCT, and CGTT) were significantly decreased in cancer patients (*Figure 6B*). Therefore, the differences in EM frequency identified by SPOT-MAS between cancer patients and healthy participants may serve as a promising target for the identification of ctDNA.

## SPOT-MAS assay combining different features of cfDNA to enhance the accuracy of cancer detection

In order to increase the sensitivity of early cancer detection while avoiding the high cost of deep sequencing, a screening test should survey a wide range of ctDNA signatures (*Moser et al., 2023*). Therefore, we utilized multiple ctDNA signatures to construct classification models for distinguishing

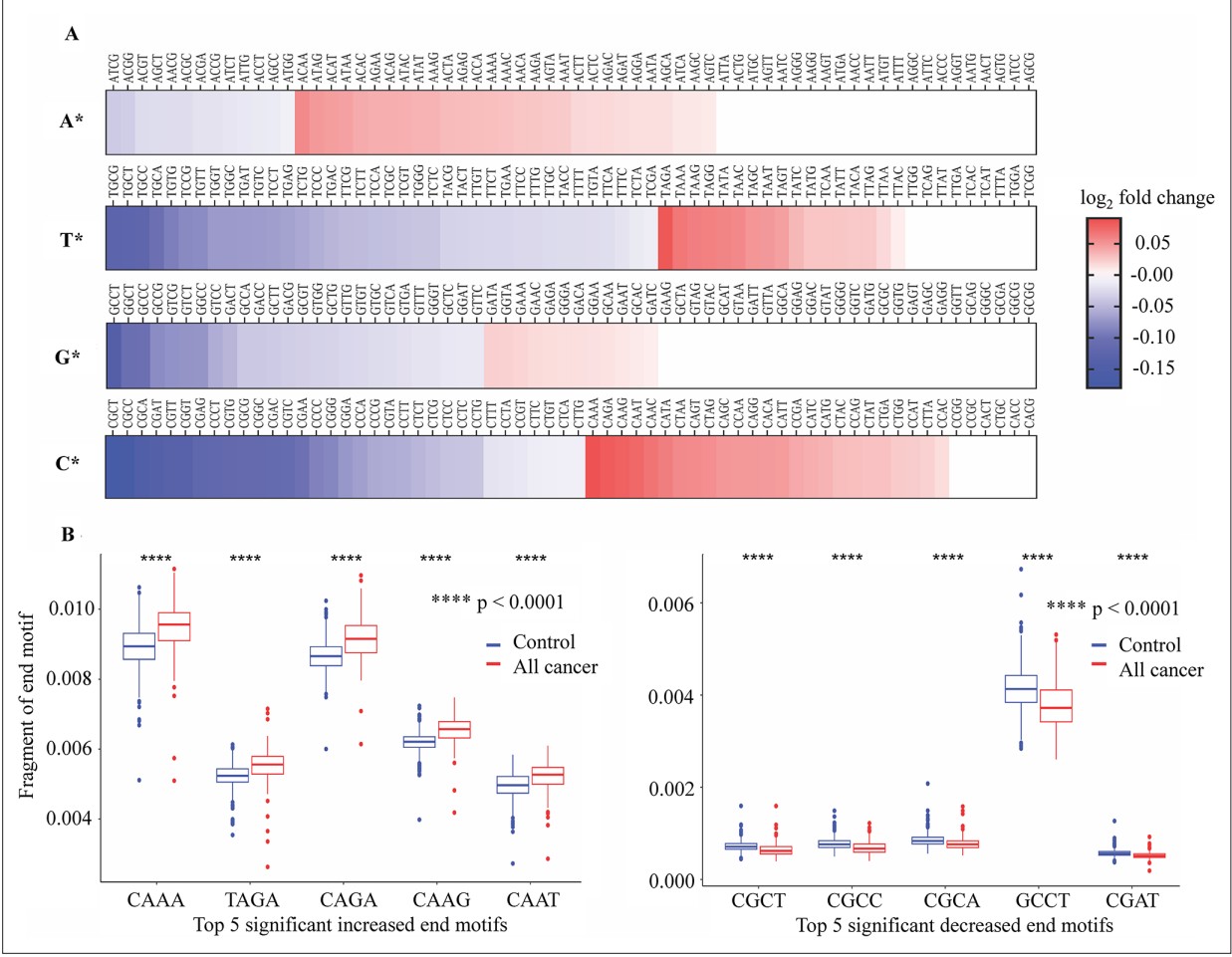

**Figure 6.** Differences in 4-mer end motif between cancer and healthy cell-free DNA (cfDNA). (**A**) Heatmap shows $\log_2$ fold change of 256 4-mer end motifs in cancer patients (*n*=499) compared to healthy controls (*n*=1076). (**B**) Box plots showing the top 10 motifs with significant differences in frequency between cancer patients (red) and healthy controls (blue) using Wilcoxon rank-sum test with Bonferroni-adjusted p-value <0.0001.

cancer patients from healthy individuals. To expand the feature space, we generated four additional features based on fragment length, including short, long, total fragment count, and short-to-long ratio, resulting in nine input feature groups (***Figure 7A***). For each feature group, we tested three different algorithms, including random forest (RF), logistic regression (LR), or extreme gradient boosting (XGB), to tune hyperparameters and select the optimal algorithms (***Figure 7A***). To evaluate the performance of these single-feature models, we performed 20-fold cross-validation on the discovery dataset and calculated 'area under the curve' (AUC) of the 'receiver operating characteristic' (ROC) curve. Among the nine features, EM-based model showed the highest AUC of 0.90 (95% CI: 0.89–0.92, ***Figure 7B***) while the SHORT-based model had the lowest AUC of 0.71 (95% CI: 0.69–0.74, ***Figure 7B***).

To assess whether combining features could improve classification, we used two strategies to construct multi-feature models. In the first strategy, all nine feature groups were concatenated into a single data frame before being fed into the RF, LR, or XGB algorithms. Of the three algorithms, the XGB model exhibited the best performance with an AUC of 0.88 (95% CI: 0.87–0.90, ***Figure 7B***). However, this AUC is still lower than that of the EM-based model (0.88 vs 0.90, ***Figure 7B***). In the second strategy, we constructed an ensemble stacking model using LR to combine the prediction results of the single-feature models. We conducted an exhaustive search approach to evaluate the performance of 511 possible combinations. The stacking ensemble model based on combining eight features, including TM, GW, CNA, FLEN, LONG, TOTAL, RATIO, and EM, exhibited the best performance and outperformed the single-feature models (***Supplementary file 1***, Table S7), with an AUC of 0.93 (95% CI: 0.92–0.95, ***Figure 7B*** and ***Figure 7—figure supplement 1***). In the independent

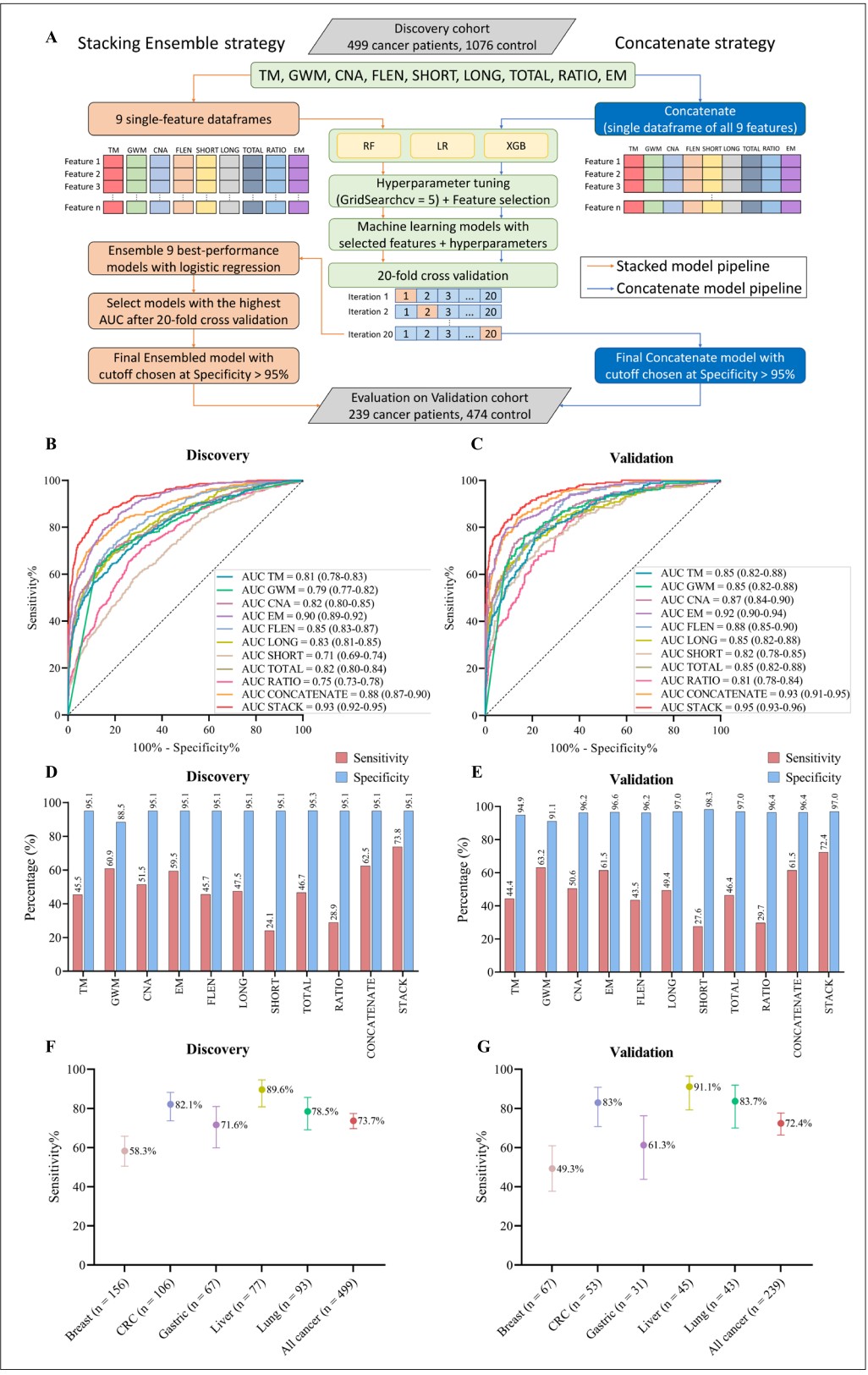

**Figure 7.** Model construction and performance validation for SPOT-MAS (screening for the presence of tumor by methylation and size). (**A**) Two-model construction strategies for cancer detection. (**B**, **C**) Receiver operating characteristic (ROC) curves comparing the performance of single-feature models, and two combination models (concatenate and ensemble stacking) in the discovery (**B**) and validation cohorts (**C**). (**D**, **E**) Bar charts showing

*Figure 7 continued on next page*

*Figure 7 continued*

the specificity and sensitivity of single-feature models and two combination models (concatenate and ensemble stacking) in the discovery (**D**) and validation cohorts (**E**). (**F**, **G**) Dot plots showing the sensitivity of SPOT-MAS assay in detection of five different cancer types in the discovery (**F**) and validation cohorts (**G**). The points and error bars represent the sensitivity and 95% confidence intervals. Feature abbreviations as follows: TM – target methylation density, GWM – genome-wide methylation density, CNA – copy number aberration, EM – 4-mer end motif, FLEN – fragment length distribution, LONG – long fragment count, SHORT – short fragment count, TOTAL – all fragment count, RATIO – ratio of short/long fragment.

The online version of this article includes the following figure supplement(s) for figure 7:

**Figure supplement 1.** Exhaustive search for the optimal stacking ensemble model.

**Figure supplement 2.** The effects of age, gender, tumor diameter, and cancer stages on model performance.

validation cohort, we obtained similar results, where the ensemble model also outperformed single-feature models, with an AUC of 0.95 (95% CI: 0.93–0.96, *Figure 7C*).

In order to ensure cost-effectiveness and minimize psychological impact of cancer screening tests in a large population, high specificity is a crucial requirement. Accordingly, we established the cutoff value for each constructed model based on a minimum specificity threshold of 95%. Of the nine single-feature models, EM and GWM models exhibited the highest sensitivities, at 59.5% and 60.9%, respectively. The stacking ensemble model achieved a sensitivity of 73.8% and a specificity of 95.1% with a cutoff value of 0.546 in the discovery cohort (*Figure 7D*), and a mean sensitivity of 72.4% and a specificity of 97.0% in the validation cohort (*Figure 7E*). Stratification of samples by cancer types revealed that the ensemble model performed most accurately in predicting liver cancer (89.6% sensitivity), followed by CRC (82.1% sensitivity), lung cancer (78.5% sensitivity), and gastric cancer (71.6% sensitivity) (*Figure 7F*, *Supplementary file 1*, Table S8). Breast cancer had the lowest detection rate of 58.3% (91/156 patients). Importantly, the performance of our ensemble model remained consistent in the validation cohort, with liver cancer again showing the highest sensitivity (91.1%), followed by lung cancer (83.7%), CRC (83.0%), gastric cancer (61.3%), and breast cancer (49.3%) (*Figure 7G*, *Supplementary file 1*, Table S8).

## Influence of clinical features on model prediction

Upon stratifying our dataset by gender, we found that there was no significant difference in the prediction of healthy status between males and females (*Figure 7—figure supplement 2*). However, in the case of cancer prediction, our model demonstrated higher accuracy in males than females in both the discovery and validation cohorts (*Figure 7—figure supplement 2*). Notably, when breast cancer samples were removed from our analysis, there was no difference in the detection rates between male and female patients (*Figure 7—figure supplement 2*), suggesting that the observed gender bias may be attributed to the high proportion of breast cancer patients (all females) in our cohort, who exhibited the lowest detection rate among the five cancer groups.

We next evaluated the potential confounding effect of age on our prediction model by examining the correlation between the model prediction scores and the participants' ages. The results revealed no significant correlation, suggesting that age differences are unlikely to affect the accuracy of our model (*Figure 7—figure supplement 2*). With regard to cancer burden (i.e. tumor size), our model performed better for cancers with higher burden, as reflected by the higher cancer scores assigned to these cases (*Figure 7—figure supplement 2*). Specifically, patients with tumor diameter ≥3.5 cm were more likely to be detected than those with a diameter <3.5 cm (*Figure 7—figure supplement 2*). Similarly, cancer stages also influence the performance of our stacking ensemble model, showing increasing detection accuracy as the stages get more advanced. In the discovery cohort, the model's accuracy was highest for stage IIIA cancers, with an AUC of 0.95 (95% CI 0.93–0.97), and lowest for stage I cancer, with an AUC of 0.90 (95% CI 0.86–0.95) (*Figure 7—figure supplement 2*). Consistently, our model performance was lower with an AUC of 0.94 (95% CI 0.89–0.98) and 0.93 (95% CI 0.90–0.96) for stage I and II cancer, respectively, increasing to 0.98 (95% CI 0.97–0.99) for stage IIIA in the validation cohort (*Figure 7—figure supplement 2*). These results demonstrated that our ensemble model can detect cancers at all stages found in our cohorts, despite a slightly lower performance in early stages (I and II) compared to non-metastatic stage (IIIA).

## SPOT-MAS enables prediction of cancer types

The ability to predict the tissue origin of ctDNA is critical for early cancer detection as this can guide subsequent diagnostic tests and treatment. Previous studies have attempted to use either fragment length or methylation landscapes to achieve this goal (*Liu et al., 2020*; *Cristiano et al., 2019*; *Klein et al., 2021*). In this study, we demonstrated the ability of SPOT-MAS to identify the TOO using low-depth bisulfite sequencing to generate multiple sets of cfDNA features. We first concatenated the nine sets of cfDNA features into a single data frame and focused our analysis on 499 cancer patients with five cancer types in the discovery cohort. We then constructed an RF and two neural network models (convolutional neural network and GCNN) to predict the TOO and used 10-fold cross-validation to estimate and compare the performance of these models (*Figure 8A* and *Figure 8—figure supplement 1*). The GCNN was chosen due to its superior performance and stability (*Figure 8—figure supplement 1* and Table S9).

We then used the GNNExplainer tool to measure the importance of different cfDNA features. Our results showed that breast cancer had the highest number of features with an important score >0.9 (497 features), while lung cancer had the lowest number of important features (126 features) (*Figure 8B*). Colorectal, gastric, and liver cancers had 363, 309, and 204 important features, respectively (*Figure 8B* and *Supplementary file 1*, Table S10). GWM and CNA were the most important features for differentiating breast, CRC, lung, gastric, and liver cancer from other cancer types, while the EM had the lowest contribution to distinguish cancer types (*Figure 8C*). Visualization of the 3D GCNN showed that this set of discriminative features could segregate the five different cancer types (*Figure 8D*), highlighting the benefits of a multimodal approach for predicting TOO.

The median accuracy for TOO identification among the five cancer types by the GCNN-based multi-feature model was 0.73 (range 0.54–0.87) in the discovery cohort (*Figure 8E*). The accuracy in the discovery cohort was highest for breast (0.87) and liver cancer (0.82) and lowest for gastric cancer (0.54). In the validation cohort, we obtained a slightly lower accuracy with a median of 0.70 (range 0.55–0.78). The accuracies for individual cancer types were 0.78 for breast, 0.76 for liver, 0.66 for colorectal, 0.63 for lung, and 0.55 for gastric cancer (*Figure 8F*). Among the five cancer types, breast cancer showed the highest TOO accuracy, possibly due to the highest number of important features detected by the model. In contrast, CRC and gastric cancer exhibited the lowest TOO accuracy with high misprediction rates between these two cancer types (0.11 and 0.19 for CRC versus gastric and gastric versus CRC, respectively). Together, our study highlights the benefits of integrating multimodal analysis with the GCNN model to capture the broad landscape of tissue-specific markers in different cancer types.

## Discussion

In an era marked by a global rise in cancer-related morbidity and mortality, the development of liquid biopsy screening tests that can detect and localize cancer at an early stage holds tremendous potential to revolutionize cancer diagnosis and therapy. However, the low amount of ctDNA fragments in plasma samples of patients with early-stage cancer as well as the molecular heterogeneity of different cancer types are known as the major challenges for liquid biopsy-based multi-cancer detection assays. Thus, sequencing at high depth coverages is required to capture enough informative cancer DNA fragments in the finite plasma sample to achieve early cancer detection. In support to this notion, many groups (*Liu et al., 2020*; *Cohen et al., 2018*; *Cristiano et al., 2019*; *Stackpole et al., 2022*) have developed assays that exploited high depth coverage of sequencing to detect ctDNA fragments in plasma of early-stage cancer patients. However, this strategy might not be cost-effective and feasible for population wide screening in developing countries. Alternatively, we argued that increasing breadth of ctDNA analysis could maximize the ability to detect ctDNA fragments with heterogeneous genetic and epigenetic changes at shallow sequencing depth, thus improving the sensitivity for multi-cancer detection. To demonstrate the feasibility of this approach, we built a stacking ensemble model to combine nine different ctDNA signatures and demonstrated its superior performance on cancer detection in comparison to single-feature models (*Figure 7B and C*). SPOT-MAS achieved a sensitivity of 72.4% at a specificity of 97.0% for detecting five common cancer types using shallow depth sequencing. Furthermore, it can predict the TOO with an accuracy of 70%.

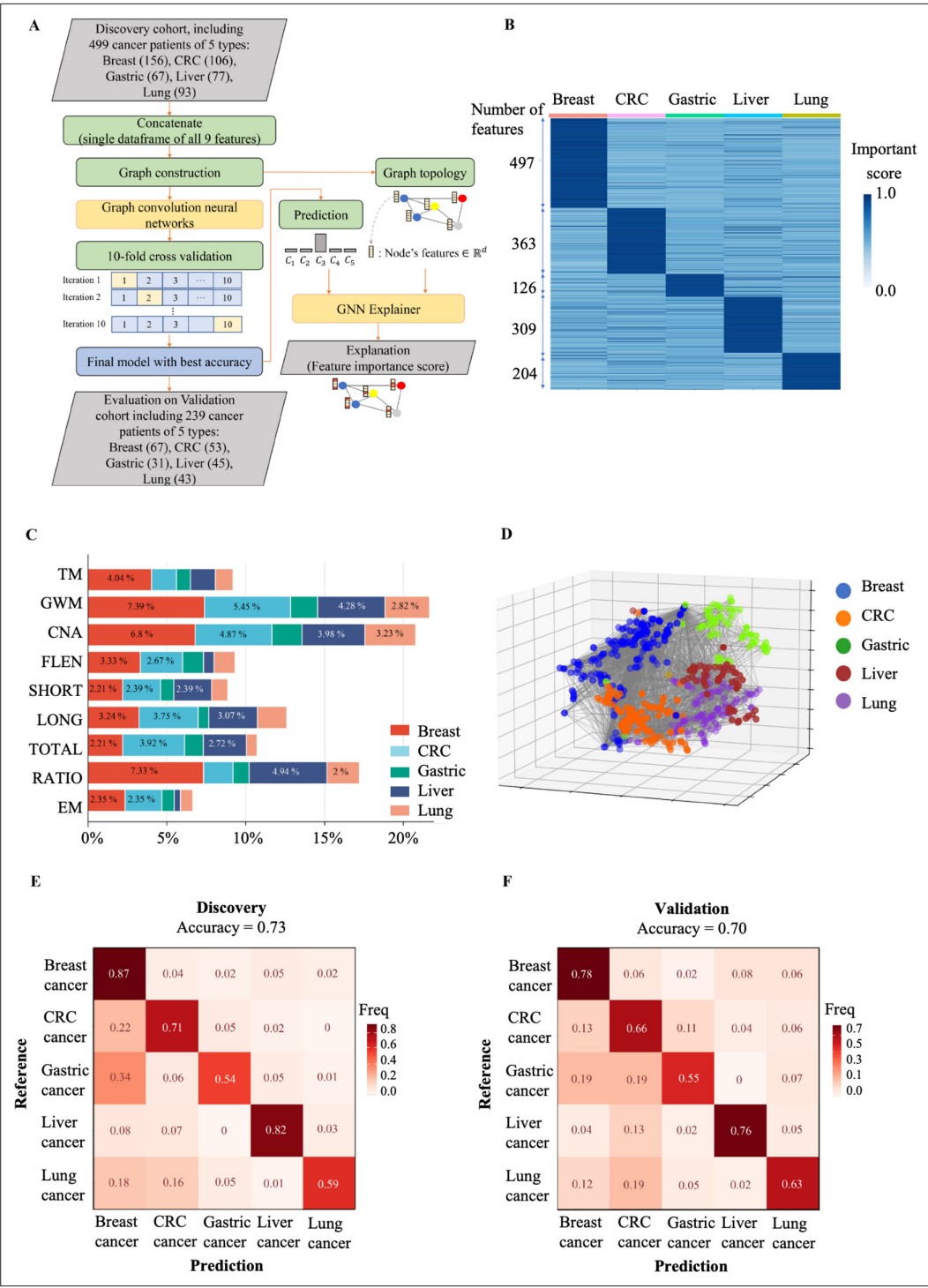

**Figure 8.** The performance of SPOT-MAS (screening for the presence of tumor by methylation and size) assay in prediction of the tissue of origin. (**A**) Model construction strategy to predict tissue of origin by combining nine sets of cell-free DNA (cfDNA) features using graph convolutional neural networks. (**B**) Heatmap shows feature important scores of five cancer types. (**C**) Bar chart indicates the contribution of important features for classifying five different cancers. (**D**) Three dimensions graph represents the classification of five cancer types. (**E**, **F**) Cross-tables show agreement between the prediction (x-axis) and the reference (y-axis) to predict tissue of origin in the discovery cohort (**E**) and validation cohort (**F**).

The online version of this article includes the following figure supplement(s) for figure 8:

**Figure supplement 1.** Construction of machine learning models for tissue of origin (TOO) identification.

*Figure 8 continued on next page*

*Figure 8 continued*

**Figure supplement 2.** Comparison of accuracy for detecting five cancer types between single-feature model and stack model.

Previous studies have reported that methylation changes at target regions could be exploited for detecting ctDNA in plasma of patients with early-stage cancer (*Xu et al., 2017*; *Luo et al., 2020*). Consistently, in TM analysis, out of 450 TM regions chosen from previous publications (*Chen et al., 2020*; *Nguyen et al., 2021*), we identified 402 regions as significant differentially methylated regions (DMRs) in cancer patients (*Figure 2A*). These DMRs were enriched for regulatory regions of well-known cancer-related gene families such as PAX family genes, TBX family genes, FOX family genes, and HOX family genes, and some have previously been reported as biomarkers for non-invasive cancer diagnosis, such as *SEPT9* and *SHOX2* (*Ilse et al., 2014*; *Warren et al., 2011*). In addition to the targeted hypermethylation regions, our study also showed widespread hypomethylation patterns across 22 autosomes of cancer patients (*Figure 3*), a hallmark of cancer (*Jones et al., 2019*). In addition to methylation alterations, recent studies have revealed that the DNA copy number, fragmentomics profile (*Cristiano et al., 2019*), and EM profile (*Jiang et al., 2020*) at genome-wide scales have been shown as useful features for healthy-cancer classification. Therefore, we propose that the combination of these markers might provide added value to increase the performance of liquid biopsy assays. We demonstrated that the same bisulfite sequencing data could be used to identify somatic CNA (*Figure 4*), cancer-associated fragment length (*Figure 5*), and EMs (*Figure 6*), highlighting the advantage of SPOT-MAS in capturing the broad landscape of ctDNA signatures without high-cost deep sequencing. For cancer-associated fragment length, we pre-processed this data into five different feature tables to better reflect the information embedded within the data. Overall, we integrated multiple features of ctDNA including methylation, fragment length, EM, and copy number changes into a multi-cancer detection model and demonstrated that this approach could distinguish healthy individuals with patients from five popular cancer types. This strategy enables increased breadth of ctDNA analysis at shallow sequencing depth to overcome the limitation of low amount of ctDNA fragments in plasma samples as well as molecular heterogeneity of cancers.

The involvement and orthogonal links of the above features in the transcriptional regulation of cancer-associated genes during carcinogenesis prompted us to examine whether the combination of multiple cancer-specific signatures in cfDNA could improve the efficiency of cancer detection (*Ulz et al., 2019*; *Ivanov et al., 2015*). We first determined the performance of models constructed using individual type of cfDNA features. Next, by performing exhaustive searches for all possible combinations of single-feature models, we identified that the stacking ensemble of seven features could achieve the AUC of 0.95 (95% CI: 0.93–0.96, *Figure 8C* and *Figure 7—figure supplement 1*), which is superior to all single-feature models. Moreover, this study showed that the feature of EM achieved the highest performance among the five examined ctDNA signatures in discriminating cancer from healthy controls (*Figure 8—figure supplement 2*). Importantly, we found that combining EM with other ctDNA signatures in a stack model could further improve the sensitivity for detecting cancer samples, with significant improvement for lung cancer patients (*Figure 8—figure supplement 2*). These findings highlighted that the multimodal analysis of multiple ctDNA signatures by SPOT-MAS could increase the breadth of ctDNA feature analysis, thus enhancing the detection sensitivity while maintaining the low cost of sample preparation and sequencing. Among the five cancer types, breast cancer showed the lowest detection rate of 58.3% and 49.3% in the discovery and validation cohort, respectively. Variations in detection rates among different cancer types have been previously reported (*Liu et al., 2020*; *Cohen et al., 2018*; *Klein et al., 2021*). Consistently, it has been reported that the detection of breast cancer, particularly in early stages, is challenging due to the low levels of ctDNA shedding and heterogeneity of molecular subtypes of breast tumors (*Liu et al., 2020*). In contrast, we obtained the highest detection rate for liver cancer patients with the sensitivity of 89.6% and 91.1% in the discovery and validation cohort, respectively. Our finding is in good agreement with the literature showing that liver tumors shed high amounts of ctDNA (*Caggiano et al., 2021*). Despite a slightly higher AUC value in the validation cohort compared to the discovery cohort, no significant differences in AUC values were observed between the two cohorts at CV of 10 or 50 (p=0.1277, DeLong's test). This result demonstrated the advantage of a multimodal approach to enhance ctDNA detection in plasma. We also conducted a survey of liquid biopsy assays to put our SPOT-MAS into the context of

current state-of-the-art in the field. *Supplementary file 1*, Table S11 showed that SPOT-MAS is using the lowest sequencing depth approach (with a depth coverage of ~0.55×) and making up for this by integrating the greatest number of cfDNA features to achieve comparable performance to other assays.

For TOO identification, our results showed that the GCNN performed the best among the models tested (*Figure 8—figure supplement 1* and *Figure 8*). GCNN has the ability to explore the similarity and mutual representation among samples, therefore achieving great success in multi-class classification tasks (*Yin et al., 2022*; *Huang and Chung, 2022*). Unlike the reference-based deconvolution approaches (*Moss et al., 2018*; *Loyfer et al., 2023*), our GCNN approach is independent of a reference methylation atlas, which was developed from tissue or cell type-specific methylation markers and thus may introduce bias due to discordance between the methylomes of tissue gDNA and plasma cfDNA (*Moser et al., 2023*; *Zhou et al., 2022a*). Although the methylation changes were reported as most predictive for TOO in previous studies (*Moss et al., 2018*; *Loyfer et al., 2023*), our results showed the contribution of each of the nine features for TOO identification (*Figure 8C*). In addition to GWM, fragment ratio (RATIO) and CNA are the major contributors to the discrimination of different tissue types. This finding provided additional evidence that the multimodal approach capturing the breadth of tissue-specific signatures could improve the accuracy of TOO identification (*Liu et al., 2020*). Our GCNN model achieved an accuracy of 0.70 for TOO prediction in validation cohort. This was comparable to the performance of CancerLocator, which was based on a probabilistic distribution model of tissue-specific methylation markers (*Kang et al., 2017*). Recently, *Liu et al., 2020*, developed a methylation atlas-based method, which achieved a higher accuracy of 93% for locating 50 types of cancer. However, this approach is based on deep genome-wide sequencing with high depth coverage of 30× (*Supplementary file 1*, Table S11), thus might not be a cost-effective approach for cancer screening in large populations, especially in low-income countries.

For an effective screening test, careful consideration of disease prevalence, cancer in this context, is imperative. Given the low prevalence of cancers, even a small proportion of false-positive test results arising from reduced assay specificity, if extrapolated to a national population, could significantly escalate the need for confirmatory imaging and biopsy procedures for benign abnormalities detected during screening. Thus, false positives can have substantial implications for both healthcare resources and patient well-being. Conversely, a screening test with high sensitivity ensures that most cancer cases are detected and minimizes delays in diagnosis. To address potential limitations posed by low sensitivity in cancer screening tests, we suggest that current liquid biopsy tests should be employed as a complementary approach to existing diagnostic methods to enhance cancer detection rates. To be used a stand-alone test, further work is required to improve its performance, with a particular emphasis on improving sensitivity while preserving high specificity.

There are several limitations in our study. First, despite using a large dataset of 738 cancer samples, there was an unequal distribution of samples among cancer types, with breast cancer accounting for 30.2% (223/738, *Supplementary file 1*, Table S2) of the total samples and gastric cancer having a much smaller representation (13.3%, *Supplementary file 1*, Table S2). As a result, our models may have been influenced by this imbalance, potentially introducing bias in the training and evaluation process. Therefore, future studies should consider incorporating more samples to better estimate the overall performance of the SPOT-MAS test. Second, tumor staging information was not available for 26.8% of cancer patients (198/738) in our study. For patients with unavailable staging information, their initial imaging examinations were conducted at the study hospitals. However, subsequent tests and surgical procedures were performed at a different hospital, as per the patients' preferences. Consequently, the original study hospitals lacked access to comprehensive tumor staging data. To address this limitation, the metastasis status of these patients was obtained via communication channels between the clinicians at the study hospitals and those at the surgery hospitals. This enabled the retrieval of limited information, adhering to an established data-sharing agreement between the two institutions. To maintain the robustness of our analysis, patients diagnosed with metastatic cancer or those with indeterminate metastatic status were subsequently excluded from the study. Therefore, all cancer patients recruited in this study were confirmed to have non-metastatic tumors. Third, the cancer patients in both the discovery and validation cohort were older than the healthy participants. Age differences could be a confounding variable of methylation and could affect the model performance (*Yusipov et al., 2020*; *Field et al., 2018*). However, we observed no significant association

between the participants' age and model prediction scores (*Figure 7—figure supplement 2*). Fourth, the ability of SPOT-MAS to differentiate cancer patients from those with benign lesions has not been examined in this study. Fifth, this study only focused on the top 5 common cancer types, thus the current version of SPOT-MAS might misidentify cancer patients of other types, resulting in lower sensitivity to real-world application. At each research sites, blood samples from both cancer patients and healthy subjects were collected in Streck Cell-Free DNA BCT tubes and subsequently transported to a central laboratory located in Medical Genetics Institute for cfDNA isolation, library preparation, and sequencing. In a recent publication (*Nguyen et al., 2023a*), we have investigated the impact of logistic time and hemolysis rates of blood samples collected from different clinical sites on cfDNA concentration and sequencing quality. We did not observe any noticeable impact of such variations on cfDNA concentrations or sequencing library yields. However, future analytical validation studies using a larger sample size are required to evaluate the impact of variation in sampling technique across different clinical sites on the robustness or accuracy of assay results. Lastly, this was a retrospective cohort study and may be biased by the nature of this study design. In an interim 6-month report of a prospective study named K-DETEK, we were encouraged by the preliminary data demonstrated the ability of SPOT-MAS to detect cancer patients who exhibited no symptoms at the time of testing (*Nguyen et al., 2023a*). Despite these promising results, the performance of SPOT-MAS as an early cancer screening test remains to be fully validated in a large, multi-center prospective study with 1–2 years of follow-up.

In conclusion, we have developed the SPOT-MAS assay to comprehensively profile methylomic, fragmentomic, CNAs, and motif end signatures of plasma cfDNA. Our large-scale case-control study demonstrated that SPOT-MAS, with its unique combination of multimodal analysis of cfDNA signatures and innovative machine-learning algorithms, can detect and localize multiple types of cancer with high accuracy at a low-cost sequencing. These findings provided important supporting evidence for the incorporation of SPOT-MAS into clinical settings as a complementary cancer screening method for at-risk populations.

## Materials and methods

### Patient enrollment

This study recruited 738 cancer patients (223 breast cancer, 159 CRC, 122 liver cancer, 136 lung cancer, 98 gastric cancer) and 1550 healthy subjects. All cancer patients were confirmed to have one of the five cancers analyzed in this study. Cancer patients were confirmed to have cancer by abnormal imaging examination and subsequent tissue biopsy confirmation of malignancy. Cancer stages were determined by the TNM (tumor, node, metastasis) system classification according to the American Joint Committee on Cancer and the International Union for Cancer Control. Our study only recruited cancer patients with non-systemic metastatic stages (stages I–IIIA) in which cancer is localized to the primary sites and has not spread to other organs. We excluded patients who were diagnosed with metastatic stage IIIB and IV cancer. All healthy subjects were confirmed to have no history of cancer at the time of enrollment. They were followed up at 6 months and 1 year after enrollment to ensure that they did not develop cancer. Study subjects were recruited from the University of Medicine and Pharmacy, Thu Duc City Hospital, Medic Medical Center, Medical Genetics Institute in Ho Chi Minh City, Vietnam, National Cancer Hospital and Hanoi Medical University in Hanoi from May 2019 to December 2022.

Written informed consent was obtained from each participant in accordance with the Declaration of Helsinki. This study was approved by the Ethics Committee of the Medic Medical Center, University of Medicine and Pharmacy and Medical Genetics Institute, Ho Chi Minh City, Vietnam. All cancer patients were treatment-naïve at the time of blood sample collection.

### Isolation of cfDNA

10 mL of blood was collected from each participant in a Cell-Free DNA BCT tube (Streck, USA). Plasma was collected from blood samples after centrifugation with two rounds (2000×*g* for 10 min and then 16,000×*g* for 10 min). The plasma fraction was aliquoted for long-term storage at –80°C. cfDNA was extracted from 1 mL plasma aliquots using the MagMAX Cell-Free DNA Isolation kit

(Thermo Fisher, USA), according to the manufacturer's instructions. Extracted cfDNA was quantified by the QuantiFluor dsDNA system (Promega, USA).

## Bisulfite conversion and library preparation

According to the manufacturer's instructions, bisulfite conversion and cfDNA purification were prepared by EZ DNA Methylation-Gold Kit (Zymo Research, D5006, USA). DNA library was prepared from bisulfite-converted DNA samples using xGen Methyl-Seq DNA Library Prep Kit (Integrated DNA Technologies, 10009824, USA) with Adaptase technology, according to the manufacturer's instructions. The QuantiFlour dsDNA system (Promega, USA) was used to analyze the concentration of DNA.

## Target region capture, whole-genome hybridization, and sequencing

DNA from library products were pooled equally, hybridized and captured using The XGen hybridization and wash kit (Integrated DNA Technologies, 1072281, USA), together with our customized panel of xGen Lockdown Probes including 450 regions across 18,000 CpG sites (Integrated DNA Technologies, USA). The construction of panel was built as previously described (*Nguyen et al., 2022a*; *Chen et al., 2020*; *Nguyen et al., 2021*). After hybridization, the flow-through product was concentrated using SpeedVac (N-Biotek, NB-503CIR, Korea) at 65°C. The samples were then added with the hybridization master mixture (hybridization buffer, hybridization enhancer, and H$_2$O) and denatured. Biotinylated P5 and P7 probes (P5-biotin: /5Biosg/AATGATACGGCGACCACCGA, P7-biotin: /5Biosg/ CAAGCAGA AGACGGCATACGAGAT) on streptavidin magnetic beads (Invitrogen, CA, USA) were hybridized with the single-stranded DNA. The captured DNA products were amplified by a PCR with free P5 and P7 primers (P5 primer: AATGATACGGCGACCACCGA, P7 primer: CAAGCAGAAGAC GGCATACGA). The concentrations of DNA libraries were determined using the QuantiFluor dsDNA system (Promega, USA). Both target and flow-through fraction were sequenced on the DNBSEQ-G400 DNA system (MGI Tech, Shenzhen, China) with 100 bp paired-end reads at a sequencing depth of 20 million reads per fraction. Data was demultiplexed by bcl2fastq (Illumina, CA, USA). FASTQ files were then examined using FastQC v. 0.11.9 and MultiQC v. 1.12.

## TM analysis

All paired-end reads were processed by Trimmomatic v 0.32 with the option HEADCROP. The trimmed reads were then aligned by Bismark v. 0.22.3. Deduplication and sorting of BAM files were conducted using Samtools v. 1.15. Reads falling into our 450 target regions were filtered using Bedtools v. 2.28. Methylation calling was performed using Bismark methylation extractor (*Nguyen et al., 2022a*). Briefly, methylation ratio was measured for each target region:

$$\text{Methylation ratio} = \frac{methylated\ cyto\sin e\ (C)}{methylated\ C + unmethylated\ C}$$

Methylation fold change from cancer to control was calculated for each target region. For analyzing differential methylated regions, significance level was set at p≤0.05, corresponding to a -log10 adjusted p-value ≥1.301 (Benjamini-Hochberg correction).

## GWM analysis

The integrated bioinformatics pipeline Methy-Pipe was used to analyze GWM. We carried out the trimming step using Trimmomatic, removing adapter sequence and low-quality bases at fragment ends (*Nguyen et al., 2022a*). The methylation ratio for each bin was calculated as follows.

$$\text{Methylation ratio} = \frac{methylated\ cyto\sin e\ (C)}{methylated\ C + unmethylated\ C}$$

Mean methylation ratio was calculated for each bin and subsequently used to plot GWM density curves. To identify bins with significant methylation changes between cancer and control group, methylation ratio in each bin of cancer samples were compared with corresponding values in control samples using Wilcoxon rank-sum test. Bins with adjusted p-value (Benjamini-Hochberg correction)≤0.05 were considered significant. Those with logFC (cancer vs control)>0 were categorized as hypermethylated bins. Those with logFC (cancer vs control)<0 were categorized as hypomethylated bins.

## CNA analysis

CNA analysis was performed using the R-package QDNAseq (*Mouliere et al., 2018*). We also used 1 Mb segmentation strategy to analyze CNA. We excluded bins that felt into the low mappability and Duke blacklist regions (*Ehrlich, 2002*). The number of reads mapped to each bin was measured by the function 'binReadCounts', and GC-content correction was conducted by the functions 'estimateCorrection' and 'correctBins'. The final CNA feature was derived by bin-wise normalizing and outlier smoothing with the functions 'normalizeBins' and 'smoothOutlierBins'. This process resulted in a feature vector of a length of 2691 bins.

To identify significant DNA gain or loss between cancer and control group, CNA values in each bin of cancer samples were compared with corresponding values in control samples using Wilcoxon rank-sum test. Bins with adjusted p-value (Benjamini-Hochberg correction)≤0.05 were considered significant. Those with logFC (cancer vs control)>0 were categorized as significant increase. Those with logFC (cancer vs control)<0 were categorized as significant decrease.

## Fragment length analysis (FLEN, SHORT, LONG, TOTAL, RATIO)

We used an in-house python script to convert the .bsalign files into BAM files and collected the fragment length from 100 to 250 bp, resulting in 151 possible fragment lengths for further analysis. The fragment frequency in each length (%) was measured by getting the proportion of reads with that length to the total read count in the range of 100–250 bp. Fragment length (bp) against fragment frequency (%) was plotted to obtain a FLEN distribution curve.

We divided the whole genome into 588 non-overlapping bins of 5 Mb (5 million bases) long and then extracted the read counts regarding these bins. Short fragments have lengths from 100 to 150 bp and long fragments have lengths from 151 to 250 bp. The ratio of short and long fragments was calculated by dividing the number of each fragment. All the short, long, and total read counts for each sample in 588 bins were normalized using z-score normalization. The short, long, and total normalized read counts and short/long ratios were chosen as features analyzed (SHORT, LONG, TOTAL, RATIO).

## EM analysis

Adaptase technology (Integrated DNA Technologies, USA) was used during library preparation to ligate adapters to ssDNA fragments in a template-independent reaction (*Jiang et al., 2020*). This step involved adding a random tail to the 5' end of reverse reads. Although median length of the tail was 8 bp and thus allowed trimming to obtain information for other analysis, the random-length tails did not allow exact determination of the 5' end of the reverse reads. Therefore, EM features were determined based on the genomic coordinate of the 5' end of the forward reads. We determined the first 4-mer sequence based on the human reference genome hg19. In 256 possible 4-mer motifs, the frequency of each motif was calculated by dividing the number of reads carrying that motif by the total number of reads, generating an EM feature vector of a length of 256 for each sample.

## Construction of machine learning models

All samples in the discovery cohort were used for model training to classify if a sample is cancerous or not. For every feature type (TM, GWM, CNA, FLEN, SHORT, LONG, TOTAL, RATIO, and EM), three machine learning algorithms, including LR, RF, and XGB, were applied. By using the 'GridSearchCV' function in the scikit-learn (v.1.0.2), model hyperparameters with the best performance were chosen with 'CV' parameter (cross-validation) set to 5. The best hyperparameters for each algorithm were found using function 'best_params_' implemented in GridSearchCV. Subsequently, feature selection was performed for each algorithm as follows: (1) for LR, the 'penalty' parameter with 'l1' (LASSO regression), 'l2' (Ridge regression), and 'none' (no penalty) were examined to select the setting with the best performance; (2) for RF and XGB, a 'SelectFromModel' function with the 'threshold' was set at 0.0001 to get all features. Then, the three algorithms (LR, RF, XGB) trained with the best hyperparameters and selected features were validated using *k*-fold cross validation approach on the dataset of training cohort with *k*-fold set to 20-fold, and 'scoring' parameter set to 'roc_auc'. This split the data into 20 groups, in which 19 groups were model-fitted and the remaining group was tested, which resulted in 20 'roc_auc' scores. The average of these scores was used to obtain the prediction performance of each model. The model with the highest 'roc_auc' average score was chosen (either LR, RF, or XGB). Ensembled models were constructed by combining probability scores of nine single-feature

base models (TM, GWM, CNA, FLEN, SHORT, LONG, TOTAL, RATIO, EM) with different combination using LR, resulting in one probability score for every sample. An extensive search was performed to evaluate the performance of all possible combinations ($n$=511) and the combination with highest AUC was selected as the final model. The model cutoff was set at the threshold specificity of >95%. This combination model performance was evaluated on an independent validation dataset to examine the model classification power.

In addition to the stacking ensemble, another combinatory strategy was examined. Instead of combining nine base models, we generated a single dataframe consisting of raw data of all nine features. The model hyperparameters tuning and features selecting were followed the same strategy as described above. After choosing the best algorithm, the model performance was also evaluated using the same external validation dataset.

## Construction of models for TOO

### Strategy 1: RF model

A single data frame of nine features in discovery cohort was used to train the RF to classify five cancer types. By using the 'GridSearchCV' function in the scikit-learn (v.1.0.2), model hyperparameters with the best performance were chosen with 'CV' parameter (cross-validation) set to 3 and 'class_weight' parameter set to 'balanced'. The best hyperparameters were found by function 'best_params_'. Then, the model was validated using $k$-fold cross-validation approach on the training cohort with $k$-fold set to 10-fold and its performance was evaluated on the validation cohort.

### Strategy 2: DNN model

Backpropagation trained the $H_2O$ deep neural network (DNN) (multi-layer feedforward artificial neural network) ($H_2O$ package, version 3.36.1.2) with stochastic gradient descent. The random grid search was selected as previously described (*Nguyen et al., 2022a*).

### Strategy 3: GCNN model

The model training utilized an input graph formed from a discovery dataset and a validation dataset as transudative setting (*Chen et al., 2020*) comprising patients diagnosed with five types of cancer: breast, colorectal (CRC), gastric, liver, and lung. The discovery dataset contains a set of sample-label pairs $\mathfrak{J} = \{(X_i, Y_i) | i = 1, \ldots, N\}$, where $X_i$ represents the $i$th sample and $Y_i$ represents $i$th label, and $N$ is the number of sample-label pairs. For each $X_i$ in the discovery dataset, a node's feature vector $f = \{F_0, \ldots, F_d\} \in R^d$ is constructed by combining groups of features, where $F_i$ is the $i$th feature, $d$ is the number of features. The same procedure was applied for the independent validation dataset. To construct an interaction graph between cancer nodes, we employed the $k$-nearest neighbors' algorithm. An interaction graph defined as $G = (V, E)$, where $V = \{X_i | i = 1, \ldots, N\}$ is a node set formed by the discovery samples, and $E = \{e_{ij}\}$ is an edge set, where $e_{ij}$ denotes an edge. Given $N$ nodes in the node set, that is $|V| = N$, a graph topology $A \in R^{N \times N}$ is defined by:

$$A_{ij} = \begin{cases} 1, & e_{ij} \in E \text{ and } d_{ij} < \delta \\ 0, & otherwise \end{cases}$$

where $d_{ij}$ is the Euclidean distance of node $i$ and $j$, and $\delta$ is set to 0.8.

In accordance with *Nguyen et al., 2021*, a GCNN was constructed for the purpose of TOO classification. The network comprised three message-passing layers, each with a hidden size of 44 and a head number of 4. TOO classification was approached as a node classification problem, wherein the model assigned each node to one of five cancer types: breast, colorectal, gastric, liver, or lung cancer. Focal loss was employed for multi-class classification optimization and the Adam optimizer was utilized for gradient-based optimization. A 10-fold cross-validation approach was implemented on the discovery dataset; nine groups were used for model training and one group for evaluation. The optimal model was selected based on its ability to achieve the highest accuracy on the validation set during 10-fold cross-validation. This model was subsequently applied to an independent validation dataset consisting of 239 cancer patients across five cancer types to obtain the performance of TOO classification.

Given the predictions of trained model and the graph topology, we estimated the feature importance score by the GNN Explainer. The feature was considered important if it satisfied:

$$F_i > \delta_f$$

where $F_i$ is the important score of $i$th feature estimated by the GNN Explainer, $\delta_f$ is the chosen cutoff and was set to 0.9.

## Statistical analysis

This study used either the Wilcoxon rank-sum test or t-test to find statistically significant differences between cancer and control. The Kolmogorov-Smirnov test was used to decide whether two cohorts have the same statistical distribution. The Benjamini-Hochberg correction was used to correct p-value for multiple comparisons (with a corrected p-value cutoff $\alpha \leq 0.05$). DeLong's test was used to compare the differences between AUCs. All statistical analyses were performed using R (4.1.0) packages, including ggplot2, pROC, and caret. 95% confident interval (95% CI) was presented in a bracket next to a value accordingly.

## Acknowledgements

We thank all participants who agreed to take part in this study, and all the clinics and hospitals who assisted in patient consultation and sample collection.

## Additional information

### Competing interests

Van Thien Chi Nguyen: VTCN is affiliated with Gene Solutions. The author has no other competing interests to declare. Trong Hieu Nguyen: HTN is affiliated with Gene Solutions. The author has no other competing interests to declare. Nhu Nhat Tan Doan: NNTD is affiliated with Gene Solutions. The author has no other competing interests to declare. Thi Mong Quynh Pham: TMQP is affiliated with Gene Solutions. The author has no other competing interests to declare. Giang Thi Huong Nguyen: GTHN is affiliated with Gene Solutions. The author has no other competing interests to declare. Thanh Dat Nguyen: TDN is affiliated with Gene Solutions. The author has no other competing interests to declare. Thuy Thi Thu Tran: TTTT is affiliated with Gene Solutions. The author has no other competing interests to declare. Thi Hue Hanh Nguyen: THHN is affiliated with Gene Solutions. The author has no other competing interests to declare. Le Anh Khoa Huynh: LAKH is affiliated with Gene Solutions. The author has no other competing interests to declare. Trung Hieu Tran: THT is affiliated with Gene Solutions. The author has no other competing interests to declare. Dac Ho Vo: DHV is affiliated with Gene Solutions. The author has no other competing interests to declare. Thi Minh Thu Tran: TMTT is affiliated with Gene Solutions. The author has no other competing interests to declare. Minh Nguyen Nguyen: MNN is affiliated with Gene Solutions. The author has no other competing interests to declare. Thi Tuong Vi Van: TTVV is affiliated with Gene Solutions. The author has no other competing interests to declare. Anh Nhu Nguyen: ANN is affiliated with Gene Solutions. The author has no other competing interests to declare. Thi Trang Tran: TTT is affiliated with Gene Solutions. The author has no other competing interests to declare. Vu Uyen Tran: VUT is affiliated with Gene Solutions. The author has no other competing interests to declare. Minh Phong Le: MPL is affiliated with Gene Solutions. The author has no other competing interests to declare. Thi Thanh Do: TTD is affiliated with Gene Solutions. The author has no other competing interests to declare. Thi Van Phan: TVP is affiliated with Gene Solutions. The author has no other competing interests to declare. Hong-Dang Luu Nguyen: HDN is affiliated with Gene Solutions. The author has no other competing interests to declare. Duy Sinh Nguyen: DSN holds equity in Gene Solutions.DSN is affiliated with Gene Solutions. The author has no other competing interests to declare. Dinh Kiet Truong: DKT is affiliated with Gene Solutions. The author has no other competing interests to declare. Hung Sang Tang: HST is affiliated with Gene Solutions. The author has no other competing interests to declare. Hoa Giang: HG holds equity in Gene Solutions. The funder Gene Solutions provided support in the form of salaries for HG who is inventor on the patent application (USPTO 17930705).HG is affiliated with Gene

Solutions. The author has no other competing interests to declare. Hoai-Nghia Nguyen: HNN holds equity in Gene Solutions. The funder Gene Solutions provided support in the form of salaries for HNN who is inventor on the patent application (USPTO 17930705).HNN is affiliated with Gene Solutions. The author has no other competing interests to declare. Minh-Duy Phan: MDP holds equity in Gene Solutions. The funder Gene Solutions provided support in the form of salaries for MDP who is inventor on the patent application (USPTO 17930705).MDP is affiliated with Gene Solutions. The author has no other competing interests to declare. Le Son Tran: LST holds equity in Gene Solutions. The funder Gene Solutions provided support in the form of salaries for LST who is inventor on the patent application (USPTO 17930705).LST is affiliated with Gene Solutions. The author has no other competing interests to declare. The other authors declare that no competing interests exist.

## Funding

| Funder | Grant reference number | Author |
|---|---|---|
| Gene Solutions | | Van Thien Chi Nguyen |
| | | Trong Hieu Nguyen |
| | | Nhu Nhat Tan Doan |
| | | Thi Mong Quynh Pham |
| | | Giang Thi Huong Nguyen |
| | | Thanh Dat Nguyen |
| | | Thuy Thi Thu Tran |
| | | Thi Hue Hanh Nguyen |
| | | Le Anh Khoa Huynh |
| | | Trung Hieu Tran |
| | | Dac Ho Vo |
| | | Thi Minh Thu Tran |
| | | Minh Nguyen Nguyen |
| | | Thi Tuong Vi Van |
| | | Anh Minh Tran |
| | | Thi Trang Tran |
| | | Vu Uyen Tran |
| | | Minh Phong Le |
| | | Thi Thanh Do |
| | | Thi Van Phan |
| | | Hong-Dang Luu Nguyen |
| | | Duy Sinh Nguyen |
| | | Hung Sang Tang |
| | | Hoa Giang |
| | | Hoai-Nghia Nguyen |
| | | Minh-Duy Phan |
| | | Le Son Tran |

The funders had no role in study design, data collection and interpretation, or the decision to submit the work for publication.

## Author contributions

Van Thien Chi Nguyen, Nhu Nhat Tan Doan, Thi Mong Quynh Pham, Thanh Dat Nguyen, Thi Hue Hanh Nguyen, Le Anh Khoa Huynh, Trung Hieu Tran, Thanh Dang Nguyen, Dac Ho Vo, Thi Minh Thu Tran, Minh Nguyen Nguyen, Thi Tuong Vi Van, Anh Nhu Nguyen, Thi Trang Tran, Vu Uyen Tran, Minh Phong Le, Thi Thanh Do, Thi Van Phan, Hong-Dang Luu Nguyen, Formal analysis; Trong Hieu Nguyen, Duy Long Vo, Thanh Hai Phan, Thanh Xuan Jasmine, Van Chu Nguyen, Huu Thinh Nguyen, Trieu Vu Nguyen, Hoa Giang, Hoai-Nghia Nguyen, Conceptualization; Giang Thi Huong Nguyen, Thuy Thi Thu Tran, Writing – original draft, Writing – review and editing; Quang Thong Dang, Patient consultancy and Screening; Thuy Nguyen Doan, Patient consultancy and Screening; Anh Minh Tran, Patient consultancy and Screening; Viet Hai Nguyen, Patient consultancy and Screening; Vu Tuan Anh Nguyen, Patient consultancy and Screening; Le Minh Quoc Ho, Patient consultancy and Screening; Quang Dat Tran, Patient consultancy and Screening; Thi Thu Thuy Pham, Patient consultancy and Screening; Tan Dat Ho, Patient consultancy and Screening; Bao Toan Nguyen, Patient consultancy and Screening; Thanh Nhan Vo Nguyen, Patient consultancy and Screening; Dung Thai Bieu Phu, Patient consultancy and Screening; Boi Hoan Huu Phan, Patient consultancy and Screening; Thi Loan Vo, Patient consultancy and Screening; Thi Huong Thoang Nai, Patient consultancy and Screening; Thuy Trang Tran, Patient consultancy and Screening; My Hoang Truong, Patient consultancy and Screening; Ngan Chau Tran, Patient consultancy and Screening; Trung Kien Le, Patient consultancy

and Screening; Thanh Huong Thi Tran, Patient consultancy and Screening; Minh Long Duong, Patient consultancy and Screening; Hoai Phuong Thi Bach, Patient consultancy and Screening; Van Vu Kim, Patient consultancy and Screening; The Anh Pham, Patient consultancy and Screening; Duc Huy Tran, Patient consultancy and Screening; Trinh Ngoc An Le, Patient consultancy and Screening; Truong Vinh Ngoc Pham, Patient consultancy and Screening; Minh Triet Le, Patient consultancy and Screening; Duy Sinh Nguyen, Patient consultancy and Screening; Van Thinh Cao, Patient consultancy and Screening; Thanh-Thuy Thi Do, Patient consultancy and Screening; Dinh Kiet Truong, Supervision; Hung Sang Tang, Patient consultancy and Screening; Minh-Duy Phan, Conceptualization, Writing – review and editing; Le Son Tran, Conceptualization, Writing – original draft, Writing – review and editing

### Author ORCIDs
Van Thien Chi Nguyen http://orcid.org/0000-0002-4449-9216
Giang Thi Huong Nguyen https://orcid.org/0000-0001-5156-4904
Huu Thinh Nguyen http://orcid.org/0000-0002-0981-7274
Quang Thong Dang http://orcid.org/0000-0003-1206-8881
Le Son Tran https://orcid.org/0000-0002-5382-3903

### Ethics
Written informed consent was obtained from each participant in accordance with the Declaration of Helsinki. This study was approved by the Ethics Committee of the Medic Medical Center, University of Medicine and Pharmacy and Medical Genetics Institute, Ho Chi Minh city, Vietnam.

Reviewer #1 (Public Review): https://doi.org/10.7554/eLife.89083.3.sa1
Reviewer #2 (Public Review): https://doi.org/10.7554/eLife.89083.3.sa2
Author Response https://doi.org/10.7554/eLife.89083.3.sa3

## Additional files

### Supplementary files
• MDAR checklist

• Supplementary file 1. Supplementary Tables.
 Table S1. Detailed clinical information of all cancer and healthy subjects. Table S2. Summary of clinical information of patients with five cancer types. Table S3. Summary of sequencing quality metrics of samples in both cohorts. Table S4. List of 450 target regions. Table S5. List of significant pathways analyzed by g:Profiler. Table S6. List of 256 4-mer end motifs (EM). Table S7. Comparison of performance between single feature-based models and stacked model. Table S8. The sensitivity of SPOT-MAS (screening for the presence of tumor by methylation and size) model for detecting different cancer types and stages at a specificity of >95%. Table S9. The accurracy of random forest (RF), deep neural network (DNN), and graph convolutional neural network (GCNN) model for tissue of origin (TOO) identification. Table S10. List of significant features for TOO identification. Table S11. Overview of liquid biopsy assays for multi-cancer early detection described in recent publications.

### Data availability
All draw data supporting the results in this paper are available at https://doi.org/10.6084/m9.figshare.24145392.v1. The code used in the study is publicly available at https://github.com/GS-ECD-Research/Multimodal_analysis_for_ECD, (copy archived at *Doan, 2023*). Sequencing data (fastqc files) from individuals are only available to interested researchers upon individual requests due to ethical restrictions imposed by The Ethic Committee of University of Medicine and Pharmacy at Ho Chi Minh City, Vietnam. To access the data, please contact the senior author Dr Le Son Tran, Medical Genetic Institute (MGI); Address: 186-188 Nguyen Duy Duong, District 10, Ho Chi Minh city, Vietnam; Email: leson1808@gmail.com. Interested researchers when requesting access to the sequencing data need to submit a data sharing agreement. The interested researchers agree to be bound by the terms and conditions in the agreement. The Director of Medical Genetics Institute will assess the data sharing agreement.

The following dataset was generated:

| Author(s) | Year | Dataset title | Dataset URL | Database and Identifier |
|-----------|------|---------------|-------------|-------------------------|
| Nguyen C | 2023 | ELIFE pan-cancer data | https://doi.org/10.6084/m9.figshare.24145392.v1 | figshare, 10.6084/m9.figshare.24145392.v1 |

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
