## [Editor Report · eLife assessment]

This study provides insights into the early detection of malignancies with noninvasive methods by developing a framework, which assesses methylation, CNA, and other genomic features. They established a **solid** model in discriminating malignancies from healthy controls, as well as the ability to distinguish tumor of origin. This **important** study will demonstrate its practical impacts in the clinic and other researchers of the field.

---

## [Referee Report · Reviewer #1 (Public Review)]

This study provides insights into the early detection of malignancies with noninvasive methods. The study contained a large sample size with an external validation cohort, which raises the credibility and universality of this model. The new model achieved high levels of AUC in discriminating malignancies from healthy controls, as well as the ability to distinguish tumor of origin. Based on these findings, prospective studies are needed to further confirm its predictive capacity.

---

## [Referee Report · Reviewer #2 (Public Review)]

The authors tried to diagnose cancers and pinpoint tissues of origin using cfDNA. To achieve this goal, they developed a framework to assess methylation, CNA, and other genomic features. They established discovery and validation cohorts for systematic assessment and successfully achieved robust prediction power.

---

## [Author Response]

The following is the authors’ response to the original reviews.

**Reviewer #1 (Public Review):**
This study provides insights into the early detection of malignancies with noninvasive methods. The study contained a large sample size with external validation cohort, which raises the credibility and universality of this model. The new model achieved high levels of AUC in discriminating malignancies from healthy controls, as well as the ability to distinguish tumor of origin. Based on these findings, prospective studies are needed to further confirm its predictive capacity.However, there are several concerns about the manuscript, which needs to be clarified or modified.1. The use of "multimodal model" will definitely increase workload of the testing. From the results of this manuscript, the integration of multimodal data did not significantly outperform the EM-based model. Is this kind of integration necessary? Is that tool really cost-effective? The authors did not convince me of its necessity, advantages, and clinical application.

To provide further evidence supporting the advantages of using multimodal model (stack model) over EM-based model, we performed the DeLong test and provided data in Table S7 and Figure S6. Our data show that the stack model outperformed the EM-based model, with significantly higher AUC (AUC difference = 0.0286, p<0.0001). Moreover, the stack model exhibited significantly higher sensitivity for detecting cancer patients of five cancer types in both discovery (73.8% versus 59.5%, p<0.0001, Figure S6A) and validation cohort (72.4% versus 61.5%, p=0.0002, Figure S6B) at comparable specificity of > 95%. The number of misclassified cases were lower when using stack model as compared to the EM-based model (Figure S6C and S6D). Strikingly, we observed that the stack model significantly improved the sensitivity for detecting lung cancer patients compared to the EM based model in both discovery (78.5% versus 44.1%, Figure S6A) and validation cohort ( 83.7% versus 55.8%, Figure S6B), indicating that other ctDNA signatures are also the important biomarkers for detecting lung cancer. Therefore, we conclude that the combination of multiple signatures of ctDNA, ie. the multimodel approach, could improve the sensitivity of multi-cancer detection.

Given the same wet lab protocol, the difference in computational time between a single EM-based model and the stack model is about 10-11 minutes per sample, but the real difference in analysis time can be reduced to ~1 min/sample by parallelization. With regards to the wet lab protocol, an important novelty of SPOT-MAS technology is its all-in-one approach that enables simultaneous analysis of different ctDNA signatures using a single blood draw and a single library reaction, greatly reducing the experimental cost. Thus, we strongly argue that our approach improves the detection sensitivity by increasing the breadth of ctDNA analysis while achieving cost effectiveness for sample preparation and sequencing with negligible trade-off of analysis time .

We have also added the following sentences in the discussion to clarify this point. (Line 618-625)

“Moreover, this study showed that the feature of EM achieved the highest performance among the five examined ctDNA signatures in discriminating cancer from healthy controls (Figure S6). Importantly, we found that combining EM with other ctDNA signatures in a stack model could further improve the sensitivity for detecting cancer samples, with significant improvement for lung cancer patients (Figure S6A and S6B). These findings highlighted that the multimodal analysis of multiple ctDNA signatures by SPOT-MAS could increase the breadth of ctDNA feature analysis, thus enhancing the detection sensitivity while maintaining the low cost of sample preparation and sequencing.”

1. The baseline characteristics of part of the enrolled patients are not clear. It seems that some of the cancer patients were diagnosed only by imaging examinations. The manuscript described "staging information was not available for 25.7% of cancer patients, who were confirmed by specialized clinicians to have non-metastatic tumors". I have no idea how did this confirmation make? According to clinicians' experience only?

Our study only recruited cancer patients with non-systemic-metastatic stages (Stage I-IIIA) in which cancer is localized to the primary sites and has not spread to other organs. We excluded patients who were diagnosed with metastatic stage IIIB and IV cancer. All healthy subjects were confirmed to have no history of cancer at the time of enrollment. They were followed up at six months and one year after enrollment. The majority of cancer patients (74.3%) were confirmed to have cancer by abnormal imaging examination and subsequent tissue biopsy confirmation of tumor staging and metastasis status. For patients with unavailable staging information (25.7%), they initially went to the study hospitals for imaging examination. Upon receiving positive imaging results (MRI scan or CT scan), they moved to another hospital for surgery, leading to missing tumor staging information at the original study hospitals. The metastasis status of these patients were later obtained via communications between the clinicians at the study hospitals and the clinicians at the surgery hospitals, subject to existing data sharing agreement between the two hospitals. For those with metastatic cancer or unclear metastatic status, they were excluded from our study.

We have added the following sentences in the method (Line 127-135) and discussion section (Line 679-688).

“Cancer patients were confirmed to have cancer by abnormal imaging examination and subsequent tissue biopsy confirmation of malignancy. Cancer stages were determined by the TNM (Tumor, Node, Metastasis) system classification according to the American Joint Committee on Cancer and the International Union for Cancer Control. Our study only recruited cancer patients with non-systemic-metastatic stages (Stage I-IIIA) in which cancer is localized to the primary sites and has not spread to other organs. We excluded patients who were diagnosed with metastatic stage IIIB and IV cancer. All healthy subjects were confirmed to have no history of cancer at the time of enrollment. They were followed up at six months and one year after enrollment to ensure that they did not develop cancer.”

“For patients with unavailable staging information, their initial imaging examinations were conducted at the study hospitals. However, subsequent tests and surgical procedures were performed at a different hospital, as per the patients' preferences. Consequently, the original study hospitals lacked access to comprehensive tumor staging data. To address this limitation, the metastasis status of these patients was obtained via communication channels between the clinicians at the study hospitals and those at the surgery hospitals. This enabled the retrieval of limited information, adhering to an established data-sharing agreement between the two institutions. To maintain the robustness of our analysis, patients diagnosed with metastatic cancer or those with indeterminate metastatic status were subsequently excluded from the study.”

1. It seems that one of the important advantages of this new model is the low depth coverage in comparing to previous screening models for cancer. The authors should discuss more on the reason why the new model could achieve comparable predictive accuracy with an obviously lower sequencing depth.

We thanked the reviewer for the suggestion. We have added the following sentences in the discussion to explain why our assay could achieve good performance at low depth sequencing. (Line 571-584)

“However, the low amount of ctDNA fragments in plasma samples of patients with early-stage cancer as well as the molecular heterogeneity of different cancer types are known as the major challenges for liquid biopsy based multi-cancer detection assays. Thus, sequencing at high depth coverages is required to capture enough informative cancer DNA fragments in the finite plasma sample to achieve early cancer detection. In support to this notion, many groups (1-4) have developed assays that exploited high depth coverage of sequencing to detect ctDNA fragments in plasma of early stage cancer patients. However, this strategy might not be cost effective and feasible for population wide screening in developing countries. Alternatively, we argued that increasing breadth of ctDNA analysis could maximize the ability to detect ctDNA fragments with heterogeneous genetic and epigenetic changes at shallow sequencing depth, thus improving the sensitivity for multicancer detection. To demonstrate the feasibility of this approach, we built a stacking ensemble model to combine nine different ctDNA signatures and demonstrated its superior performance on cancer detection in comparison to single-feature models (Figure 7B and 7C).”

1. The readability of this manuscript needs to be improved. The focus of the background section is not clear, with too much detail of other studies and few purposeful summaries. You need to explain the goals and clinical significance of your study. In addition, the results section is too long, and needs to be shortened and simplified. Move some of the inessential results and sentences to supplementary materials or methods.

We thank the reviewer for these constructive suggestions. Accrodingly, we have reduced the details of other studies (Line 85-91) as follows:

“In recent years, there has been considerable interest in exploring the potential of ctDNA alterations for early detection of cancer (5, 6). One such approach is the PanSeer test, which uses 477 differentially methylated regions (DMRs) in ctDNA to detect five different types of cancer up to four years prior to conventional diagnosis (7). The DELFI assay employs a genome-wide analysis of ctDNA fragment profiles to increase sensitivity in early detection (1). Recently, the Galleri test has emerged as a multi-cancer detection assay that analyses more than 100,000 methylation regions in the genome to detect over 50 cancer types and localize the tumor site (8).”

We have modified the text in the introduction to explain the goals and clinical significance of our study (Line 111-123)

“In this study, we aimed to expand our multimodal approach, SPOT-MAS, to comprehensively analyze methylomics, fragmentomics, DNA copy number and end motifs of cfDNA and evaluate its utility to simultaneously detecting and locating cancer from a single screening test.”“Our findings demonstrate that the multimodal approach of SPOT-MAS enables profiling of multiple ctDNA signatures across the entire genome at low sequencing depth to detect five different cancer types in their early stages. Beyond detecting the presence of cancer signals, our assay was able to predict the tumor location, which is important for clinicians to fast-track the follow-up diagnostic and guide necessary treatment. Thus, SPOT-MAS has the potential to become a universal, simple, and cost-effective approach for early multi-cancer detection in a large population.”

**Reviewer #2 (Public Review):**
The authors tried to diagnose cancers and pinpoint tissues of origin using cfDNA. To achieve the goal, they developed a framework to assess methylation, CNA, and other genomic features. They established discovery and validation cohorts for systematic assessment and successfully achieved robust prediction power.1. Still, there are places for improvement. The diagnostic effect can be maximized if their framework works well in early-stage cancer patients. According to Table 1, about 10% of the participants are stage I. Do these cancers also perform well as compared to late stage cancers?

We have performed the comparison of SPOT-MAS performance on different stages and provided the data in Supplementary table S8 and Supplementary Figure S4J and S4L. Our data showed that SPOT-MAS achieved lower sensitivity for detecting stage I and II cancers as compared to stage IIIA cancers in both discovery (61.54% and 69.82% for stage I and II respectively versus 78.67% for stage IIIA, Supplementary table 8) and validation cohort (73.91% and 62.32% for stage I and II, respectively versus 88.31% for stage IIIA, Supplementary table 8). This suggested that cancer stages can influence the performance of our models.

1. Can authors show a systematic comparison of their method to other previous methods to summarize what their algorithm can achieve compared to others.

We have conducted a systematic comparison of our method with others in the Supplementary Table S11.

**Reviewer #1 (Recommendations For The Authors):**
There are still points for the authors to clarify and consider for incorporation into revision.Please first clarify the issues mentioned in "public review". Several complements are needed.

We have addressed all of the reviewer’s comments in “public review”.

1. Line 72-73: Different approaches of early cancer screening assays have different features, application scenarios, and of course, limitations. It's too vague to describe in this way. More importantly, diagnosis of malignancies relies on pathological diagnosis, I don't think the results of unsuccessful screening would be overdiagnosis and overtreatment. That's overstatements.

We have rewritten the statement as follows (Line 72-75)

“Although currently guided screening tests have each been shown to provide better treatment outcomes and reduce cancer mortality, some of them are invasive, thus having low accessibility. Importantly, most of them are single cancer screening tests, which may result in high false positive rates when used sequentially.”

1. Line 115-130: The findings in this study shouldn't be introduced here.

We have removed this section.

1. Line 496-498: It surprised me that the model performed even better in independent validation cohort, which is quite different from the usual situations. Please explain it.

We agree with the reviewer that model performance in independent validation cohort is often lower than in discovery cohort. In our case, we have carefully confirmed our data by utilizing cross-validation (CV). Cross-validation is a widely used process in which the data being used for training the model is separated into folds or partitions and the model is trained and validated for each fold; the performance estimates are then calculated to obtain mean and confidence interval (GraphPad Prism, Wilson/Brown method). To further confirm our findings, we have increased the cross-validation fold into 50, and consistently detected no significant difference in the performance between Discovery and Validation cohorts (p=0.1277, DeLong’s test).

We have added the following sentence in the discussion to explain this (Line 633-635)

“Despite a slightly higher AUC value in the validation cohort compared to the discovery cohort, no significant differences in AUC values were observed between the two cohorts at CV of 10 or 50 (p=0.1277, DeLong’s test).”

1. Line 499-501: For the cut-off value selection, the authors thought that for cancer screening, specificity is more important than sensitivity? It's controversial. The sensitivity is only approximately 70%, I think that a missed diagnosis is even worse.

We agree with the reviewer that both specificity and sensitivity are important metrics of a cancer detection test. However, there is a trade-off between sensitivity and specificity and the preference for either one of them remains a controversial topic. For a screening test, the preference should be determined by considering the prevalence of the disease, in this case - cancer. The low prevalence of cancers indicates that even a small percentage of false-positive test results due to low specificity of the assay, spread across a national population, would hugely increase the demand for confirmatory imaging as well as biopsy sampling of imaging-detected benign abnormalities (9). Thus, false positives have obvious implications for health-care resources as well as patient well-being. Conversely, higher sensitivities will make sure that more cancer cases are detected and avoid delays in diagnosis. To mitigate the impact of insufficient sensitivity of a cancer screening test, it is important to consult the test-takers that current liquid biopsy tests should only be used as a complementary approach to the available diagnosis tests to increase rates of cancer detection. To be used as a stand-alone test, further work is required to improve its performance, with more focus on increasing sensitivity while maintaining high specificity.

We have added the following sentences in the discussion to explain why we set a high threshold of specificity (Line 660-671)

“For an effective screening test, careful consideration of disease prevalence, cancer in this context, is imperative. Given the low prevalence of cancers, even a small proportion of false-positive test results arising from reduced assay specificity, if extrapolated to a national population, could significantly escalate the need for confirmatory imaging and biopsy procedures for benign abnormalities detected during screening. Thus, false-positives can have substantial implications for both healthcare resources and patient well-being. Conversely, a screening test with high sensitivity ensures that most cancer cases are detected and minimizes delays in diagnosis. To address potential limitations posed by low sensitivity in cancer screening tests, we suggest that current liquid biopsy tests should be employed as a complementary approach to existing diagnostic methods to enhance cancer detection rates. To be used a stand-alone test, further work is required to improve its performance, with a particular emphasis on improving sensitivity while preserving high specificity.”

1. The methylation profiles have been used broadly in ctDNA, while your also integrated the fragmentomics, copy number aberration and end motif into the new model. In the discussion section, it would be better to further compare your new model with several previous models based on conventional ctDNA methylation markers (10, 11) for early detection of malignancies. What are the advantages of adding the other two types of data? Why the new model could achieve comparable predictive accuracy with an obviously lower sequencing depth?

We thank the reviewer for the suggestion. We have added the following sentences in the discussion to highlight the novelty of our multimodal approach. (Line 587-610)

“Previous studies have reported that methylation changes at target regions could be exploited for detecting ctDNA in plasma of patients with early-stage cancer (10, 11).”

“In addition to methylation alterations, recent studies have revealed that the DNA copy number, fragmentomics profile (1) and end motif profile (12) at genome wide scales have been shown as useful features for healthy-cancer classification. Therefore, we propose that the combination of these markers might provide added value to increase the performance of liquid biopsy assays. We demonstrated that the same bisulfite sequencing data could be used to identify somatic CNA (Figure 4), cancer-associated fragment length (Figure 5) and end motifs (Figure 6), highlighting the advantage of SPOT-MAS in capturing the broad landscape of ctDNA signatures without high cost deep sequencing. For cancer-associated fragment length, we pre-processed this data into five different feature tables to better reflect the information embedded within the data. Overall, we integrated multiple features of ctDNA including methylation, fragment length, end motif and copy number changes into a multi-cancer detection model and demonstrated that this approach could distinguish healthy individuals with patients from five popular cancer types. This strategy enables increased breadth of ctDNA analysis at shallow sequencing depth to overcome the limitation of low amount of ctDNA fragments in plasma samples as well as molecular heterogeneity of cancers.”

Moreover, we have conducted a systematic comparison of our method with others in the Supplementary Table 11.

1. Line 667-668: The wording should be modest. "Successfully detect and localize" is not appropriate.

We have rewritten the sentence. (Line 713-716)

“Our large-scale case-control study demonstrated that SPOT-MAS, with its unique combination of multimodal analysis of cfDNA signatures and innovative machine-learning algorithms, can detect and localize multiple types of cancer with high accuracy at a low-cost sequencing.”

**Reviewer #2 (Recommendations For The Authors):**
1. Are the patients and controls all from Vietnam? If I am not mistaken, it is hard to find demographic information for controls. Also it is not clear if samples from controls were processed simultaneously or at a same institution or using the same protocol etc.

We thank the reviewer for asking this question. All cancer patients and controls are from Vietnam, who were recruited from five hospitals including Medic Medical Center, University Medical Center Ho Chi Minh City, Thu Duc City Hospital, National Cancer Hospital and Hanoi Medical University. At each research sites, blood samples from both cancer patients and healthy subjects were collected in in Streck Cell-Free DNA BCT tubes and subsequently transported to a central laboratory located in Medical Genetics Institute for cfDNA isolation, library preparation and sequencing. In a recent publication (10), we have investigated the impact of logistic time and hemolysis rates of blood samples collected from different clinical sites on cfDNA concentration and sequencing quality. We did not observe any noticeable impact of such variations on cfDNA concentrations or sequencing library yields. However, future analytical validation studies are required to evaluate the impact of variation in sampling technique across different clinical sites on the robustness or accuracy of assay results.

We have added the following sentences in the discussion to highlight this important point (Line 696-704)

“At each research sites, blood samples from both cancer patients and healthy subjects were collected in in Streck Cell-Free DNA BCT tubes and subsequently transported to a central laboratory located in Medical Genetics Institute for cfDNA isolation, library preparation and sequencing. In a recent publication (10), we have investigated the impact of logistic time and hemolysis rates of blood samples collected from different clinical sites on cfDNA concentration and sequencing quality. We did not observe any noticeable impact of such variations on cfDNA concentrations or sequencing library yields. However, future analytical validation studies using a larger sample size are required to evaluate the impact of variation in sampling technique across different clinical sites on the robustness or accuracy of assay results.”

References

1. Cristiano S, Leal A, Phallen J, Fiksel J, Adleff V, Bruhm DC, et al. Genome-wide cell-free DNA fragmentation in patients with cancer. Nature. 2019;570(7761):385-9.

2. Cohen JD, Li L, Wang Y, Thoburn C, Afsari B, Danilova L, et al. Detection and localization of surgically resectable cancers with a multi-analyte blood test. Science. 2018;359(6378):926-30.

3. Liu MC, Oxnard GR, Klein EA, Swanton C, Seiden MV. Sensitive and specific multi-cancer detection and localization using methylation signatures in cell-free DNA. Ann Oncol. 2020;31(6):745-59.

4. Stackpole ML, Zeng W, Li S, Liu C-C, Zhou Y, He S, et al. Cost-effective methylome sequencing of cell-free DNA for accurately detecting and locating cancer. Nature Communications. 2022;13(1):5566.

5. Constantin N, Sina AA, Korbie D, Trau M. Opportunities for Early Cancer Detection: The Rise of ctDNA Methylation-Based Pan-Cancer Screening Technologies. Epigenomes. 2022;6(1).

6. Phan TH, Chi Nguyen VT, Thi Pham TT, Nguyen VC, Ho TD, Quynh Pham TM, et al. Circulating DNA methylation profile improves the accuracy of serum biomarkers for the detection of nonmetastatic hepatocellular carcinoma. Future Oncol. 2022;18(39):4399-413.

7. Chen X, Gole J, Gore A, He Q, Lu M, Min J, et al. Non-invasive early detection of cancer four years before conventional diagnosis using a blood test. Nature Communications. 2020;11(1):3475.

8. Jamshidi A, Liu MC, Klein EA, Venn O, Hubbell E, Beausang JF, et al. Evaluation of cell-free DNA approaches for multi-cancer early detection. Cancer Cell. 2022;40(12):1537-49.e12.

9. Ignatiadis M, Sledge GW, Jeffrey SS. Liquid biopsy enters the clinic - implementation issues and future challenges. Nat Rev Clin Oncol. 2021;18(5):297-312.

10. Xu RH, Wei W, Krawczyk M, Wang W, Luo H, Flagg K, et al. Circulating tumour DNA methylation markers for diagnosis and prognosis of hepatocellular carcinoma. Nat Mater. 2017;16(11):1155-61.

11. Luo H, Zhao Q, Wei W, Zheng L, Yi S, Li G, et al. Circulating tumor DNA methylation profiles enable early diagnosis, prognosis prediction, and screening for colorectal cancer. Sci Transl Med. 2020;12(524).

12. Jiang P, Sun K, Peng W, Cheng SH, Ni M, Yeung PC, et al. Plasma DNA End-Motif Profiling as a Fragmentomic Marker in Cancer, Pregnancy, and Transplantation. Cancer Discovery. 2020;10(5):664-73.